# Activation function cyclically switchable convolutional neural network model

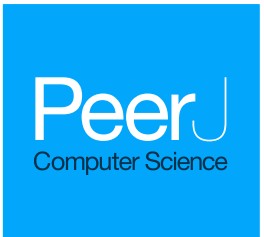

İsmail Akgül

Department of Computer Engineering, Erzincan Binali Yıldırım University, Erzincan, Turkey

## ABSTRACT

Neural networks are a state-of-the-art approach that performs well for many tasks. The activation function (AF) is an important hyperparameter that creates an output against the coming inputs to the neural network model. AF significantly affects the training and performance of the neural network model. Therefore, selecting the most optimal AF for processing input data in neural networks is important. Determining the optimal AF is often a difficult task. To overcome this difficulty, studies on trainable AFs have been carried out in the literature in recent years. This study presents a different approach apart from fixed or trainable AF approaches. For this purpose, the activation function cyclically switchable convolutional neural network (AFCS-CNN) model structure is proposed. The AFCS-CNN model structure does not use a fixed AF value during training. It is designed in a self-regulating model structure by switching the AF during model training. The proposed model structure is based on the logic of starting training with the most optimal AF selection among many AFs and cyclically selecting the next most optimal AF depending on the performance decrease during neural network training. Any convolutional neural network (CNN) model can be easily used in the proposed model structure. In this way, a simple but effective perspective has been presented. In this study, first, ablation studies have been carried out using the Cifar-10 dataset to determine the CNN models to be used in the AFCS-CNN model structure and the specific hyperparameters of the proposed model structure. After the models and hyperparameters were determined, expansion experiments were carried out using different datasets with the proposed model structure. The results showed that the AFCS-CNN model structure achieved state-of-the-art success in many CNN models and different datasets.

## INTRODUCTION

Neural networks are one of the modeling methods that play an important role in the fields of machine learning and deep learning (*Goel, Goel & Kumar, 2023*). The basic building blocks of these networks are artificial neurons inspired by biological nerve cells (*Montesinos López, Montesinos López & Crossa, 2022*; *Liu et al., 2023*). Artificial neurons that mimic biological neuron functions (*Cao et al., 2022*) apply a mathematical function to process the inputs they receive and produce output (*Han et al., 2022*; *Gülmez, 2023*; *Yevick, 2024*). In this process, the flow of information between neurons and the responses of

Corresponding author
İsmail Akgül, iakgul@erzincan.edu.tr

neurons to inputs are largely regulated through activation function (AF) (*Vijayaprabakaran & Sathiyamurthy, 2022*). AF used in these processes is a critical element that determines how the neuron will respond to the signals it receives.

AFs directly affect the training and performance of the model as they provide an output based on the inputs in the neural network (*Akgül, 2023*). They are divided into two main categories: linear and nonlinear functions. While linear AFs enable the model to perform well on simpler problems, more complex functions are needed to model nonlinear relationships (*Vargas et al., 2021*). The task of AF is to learn abstract properties of data through nonlinear modification (*Kalim, Chug & Singh, 2024*). That is, to help learn nonlinear and complex mappings between inputs and their corresponding outputs (*Sharma, Sharma & Athaiya, 2020*; *Subramanian et al., 2024*). Recent research has found that AF plays an important role in introducing nonlinearity to improve the performance of deep learning networks (*Rajanand & Singh, 2024*). This nonlinear ability of AF has brought true artificial intelligence to deep neural networks (*Wang, Ren & Wang, 2022*).

Many AFs have been proposed throughout the history of machine learning. Some of the AF types commonly used in neural networks are: Binary Step, Linear, Sigmoid, Hyperbolic Tangent (Tanh), Rectified Linear Unit (ReLU), Leaky ReLU, Parametric RELU (PReLU), Exponential Linear Units (ELU), SELU, Softmax, Mish, Swish/Sigmoid Linear Unit (SiLU), Gaussian Error Linear Unit (GELU), Logish, Softplus and Softsign (*Yuen et al., 2021*; *Wang, Ren & Wang, 2022*; *Kiaei et al., 2023*; *Kalim, Chug & Singh, 2024*; *Verma, Chug & Singh, 2024*). AFs such as Sigmoid, Tanh, ReLU, and Leaky ReLU are the most widely used nonlinear functions in neural networks. Each function responds differently to inputs and offers specific advantages for specific problems. Sigmoid and Tanh AFs are both s-shaped and are used to classify objects where the output is limited to {0, 1} to {−1, 1} respectively (*Verma, Chug & Singh, 2024*). ReLU AF is used for quantification, classification, and reinforcement learning problems. Leaky ReLU, ELU, and Softplus AFs are part of the ReLU AF family. Leaky ReLU AF is a version of ReLU with a non-zero gradient to prevent the gradient from reaching zero. Elu and Softplus AFs solve the problem by creating smoothness and continuity in the environment. Newer AFs such as Mish, Swish/SiLU, and GELU have built-in regularization to prevent over-fitting of models (*Yuen et al., 2021*).

The purpose of machine learning algorithms is to determine the most optimal model to solve a specific problem. To determine the most optimal model, it is necessary to select the most optimal AF, like many parameters (*Yuen et al., 2021*). However, selecting the most optimal AF to train the neural network is not always an easy task (*Kiaei et al., 2023*). Since linear AFs limit the learning performance of deep learning models, nonlinear AFs are mostly preferred. Nonlinear AFs can be classified as fixed-parameter and trainable/adaptive AFs, depending on whether the AF parameter is fixed or modified during the training process of deep learning models (*Kiliçarslan & Celik, 2024*). Although the effectiveness of trainable or adaptive AFs has been examined in areas with abundant data, such as image classification problems, significant gaps remain in understanding their impact on classification accuracy and prediction uncertainty in environments characterized by limited data availability (*Pourkamali-Anaraki et al., 2024*).

Although fixed parameter AFs have been widely used in image classification models in previous studies, they have important shortcomings such as low classification accuracy. In recent years, adding parameters to AF to improve its performance has become a popular research topic. Excellent progress has been made with adaptive AFs obtained in this way (*Jiang, Xie & Zhang, 2022*). Many recent studies have been conducted on trainable or adaptive AF in the literature, especially in recent years (see 'Related Works'). In these studies, a parametric variation of an existing AF was created by adding new parameters to commonly used AF types. Some of these created parametric AFs change the activation in the positive domain (*Vargas et al., 2021*). Another part focuses on either assigning a different slope to the negative domain of AF or converting the negative value to a positive value instead of zero (*Jiang, Xie & Zhang, 2022*). Although cyclic switching strategies such as cyclic learning rates (*Smith, 2017*) and cyclic precision (*Fu et al., 2021*) have been discussed during training, the cyclic AF switching strategy has not been discussed. In the studies, new AFs were generally proposed by adding an adaptive parameter to the AF, and the AF was used without a switch throughout the training. However, no study has been found on switching AF during training. In this study, unlike other studies, instead of an AF proposal, the ability to switch the AF with another AF during neural network training was adopted. The AFs to be used during training can be either fixed-parameter or adaptive. Thus, many AF switches are allowed during training (while training continues). In summary, it can be used in many AF neural networks thanks to the instantaneous AF switch while neural network training continues.

In this study, the activation function cyclically switchable convolutional neural network (AFCS-CNN) model structure was proposed, which allows cyclical switching of AF, which has an important place in neural networks. A first was achieved by proposing a new model from a unique perspective in the literature. Many experiments have been applied to determine the effectiveness of the proposed model. After a series of ablation experiments were carried out on the Cifar-10 dataset, expansion experiments were performed on the V2 Plant Seedling and APTOS 2019 Blindness Detection datasets, and the performance of the proposed model was measured. The highlight of this study to the literature can be summarized as follows:

1) A new model structure called AFCS-CNN has been proposed, which enables cyclical switching of AF.
2) Unlike the studies in the literature, instead of an AF proposal, the ability to switch the AF with another AF during neural network training has been adopted.
3) The concept of cyclic AF switching strategy during model training has been introduced.
4) A first was achieved by designing a model structure that had not been tried before, thanks to instant AF switches during neural network training.
5) The algorithm of the proposed model structure is designed to allow easy integration of all convolutional neural network (CNN) models into the structure.
6) Training with the proposed model structure has provided superior success in many problems compared to training with fixed AFs.

7) A state-of-the-art success has been achieved with the proposed model structure in plant seedling classification.

The remaining part of the study is organized as follows. Recent studies on AF development can be found in the 'Related Works', dataset preprocessing, training parameters, performance metrics and model structure are noted in 'Materials and Methods', performance and results are discussed in 'Results and Discussions', and the article concludes in the 'Conclusion'.

## RELATED WORKS

In this section, recent studies on the development of trainable or adaptable AFs are comprehensively reviewed.

In a study on trainable or adaptive AFs, a trainable AF named Modified Mexican ReLU (MMeLU) was proposed to solve the complexity problem and improve the model performance. A fully Bayesian model is developed to automatically estimate both model weights and AF parameters from the training data. The proposed method is designed to learn the recommended AF weights and parameters directly from the data without any user configuration. The proposed method has been tested on various datasets covering both classification and regression tasks and implemented on various CNN architectures. The results showed that the proposed approach is useful in improving the model accuracy due to the proposed AF and Bayesian estimation of the parameters (*Fakhfakh & Chaari, 2024*). In another study, a new AF named ErfReLU based on error function (erf) and ReLU was developed. The performance of the proposed function is compared with nine different trainable AFs. CIFAR-10, MNIST, and FMNIST benchmark datasets were used to evaluate the effectiveness of AFs with MobileNet, VGG16, and ResNet models. The proposed AF has achieved better accuracy than other state-of-the-art adaptive AFs (*Rajanand & Singh, 2024*).

In another study, a novel trainable AF, adaptive piecewise approximated activation linear unit (APALU), was proposed to improve the learning performance of deep learning on a wide range of tasks. It has been stated that the proposed function exhibits significant improvements over AFs commonly used in image classification, anomaly detection, sign language recognition, and regression tasks (*Subramanian et al., 2024*). In another study, two trainable Gaussian-based AFs were proposed for sensor-based human activity recognition (HAR). Experiments have been conducted on Opportunity and UniMiB SHAR benchmark datasets with the proposed Four-Parameter Activation Gaussian Radial Basis Function (T4GRBF) and Weighted Gaussian Radial Basis Function (WGRBF). On both datasets, the results showed that Trainable Gaussian-based AFs fit the training data better and faster than standard AFs (*Machacuay & Quinde, 2024*). In another study, a generalized activation function called Generalized Exponential Parametric Activation Function (GEPAF) was proposed. In applications on real-world supply chain datasets, the proposed function outperformed popular AFs while demonstrating at least a 30% improvement in regression evaluation metrics and better loss reduction properties (*Attarde & Sayyad, 2024*).

In another study, PolySigmoid, PolySoftplus, PolyGeLU, PolySwish, and PolyMish polynomial versions of existing shape-shifting AFs such as Sigmoid, Softplus, GeLU, Swish, and Mish were proposed. As a result of the accuracy comparisons, it was determined that the performance of the polynomial versions of AFs was similar to or better than their counterparts (*Herrera-Alcántara & Arellano-Balderas, 2024*). Another study proposed an adaptive ReLU based on the genetic algorithm (GA) profile to improve the restrictive ability of the original ReLU and determine the best threshold value to allow adaptation-based neuron activation. As a result of experiments for breast cancer classification, it was found that the proposed method showed improved accuracy and improved classification performance from 95.0% to 98.5% compared to other well-known AFs (*Razali et al., 2024*).

Another study examined the effectiveness of trainable (non-periodic) AFs for neural implicit k-space (NIK) in the context of non-Cartesian Cardiac MRI. The proposed NIKs with trainable AFs are evaluated on 42 radially sampled datasets from six subjects and qualitatively and quantitatively outperform other state-of-the-art reconstruction methods including NIK with fixed periodic AFs (*Haft et al., 2024*). In another study, a new transformative adaptive activation function (TAAF) that allows any vertical and horizontal translation and scaling was investigated. It has been emphasized that TAAFs generalize more than 50 existing AFs and use similar concepts to more than 70 other AFs (*Kunc, 2024*).

In another work, a unified adaptive AF called adaptive activation (AdAct) was presented, which aims to improve neural network performance by adaptively selecting and combining the most suitable AFs for different tasks and datasets. The effectiveness of ReLU and its variants, including ELU, LReLU, PReLU, RReLU, and newer functions such as Swish and Mish, has been investigated by integrating them into the AdAct function. In the study where ConvNet variants were used on FMNIST, CIFAR10, SVHN, and FER datasets, AdAct showed better performance than other AFs (*Maiti, 2024*). In another study, in which the adaptive activation algorithm AdAct was proposed, it was emphasized that there were improvements in the performance of feedforward and convolutional neural networks (*Rane et al., 2024*). In another study, a new model, the expressive neural network (ENN), was presented in which nonlinear AFs were modeled using discrete cosine transform (DCT) and adapted using backpropagation during training. ENN has outperformed state-of-the-art non-adaptive AFs by adapting AFs for classification and regression problems (*Martinez-Gost, Pérez-Neira & Lagunas, 2024*).

In another study, a novel modulation window radial basis function neural network (MW-RBFNN) with a tunable AF was proposed. A raised cosine radial basis function (RC-RBF) was implemented as a shape-tunable AF of MW-RBFNN by adaptively modulating it by an exponential function. Simulation results of many different application cases have shown that this proposed MW-RBFNN is effective (*Lin et al., 2024*). In another study, a new adaptive deep neural networks (ADN) technique is proposed for partial differential equations (PDE) with corner singular solutions. Three different adaptive techniques were used for this, such as an adaptive loss function, an adaptive activation function, and an adaptive sampling strategy. The results showed that ADN achieved higher accuracy and

faster convergence speed compared to some traditional DNN methods (*Zeng, Liang & Zhang, 2024*).

In a different study, it was studied that a customized adaptive activation function (AAF) can match the accuracy of a deep neural network (DNN). A field programmable gate arrays (FPGA) hardware implementation for a customized segmented spline curve neural network (SSCNN) structure is designed to replace the traditional fixed AF with an AAF. The proposed SSCNN implementation achieved similar accuracy using 40% less hardware resources and block RAM compared to DNN (*Jiang et al., 2024*). In a similar study, a local active memristor with a conductance function value range limited between 0 and 1 was designed to create an adaptive AF called memristive PReLU (MPReLU). A heterogeneous memristive Hopfield neural network with neurons using different activation functions was presented and a hardware implementation of this neural network with adaptive AF was designed. The experimental results largely coincided with those obtained using numerical simulations (*Wang, Liang & Deng, 2024*).

## MATERIALS AND METHODS

### Dataset and preprocessing

In this study, the Cifar-10 (*Krizhevsky, 2009*) dataset was used to determine the CNN model to be used in the AFCS-CNN model structure and the special hyperparameters of AFCS-CNN, and V2 Plant Seedling (*Kaggle, 2024*; *Giselsson et al., 2017*) and APTOS 2019 Blindness Detection (*APTOS 2019, 2024*) datasets were used to test the model performance. Cifar-10 dataset consists of 10 classes and each of the 60,000 images in this dataset has dimensions of $32 \times 32 \times 3$ pixels. The V2 Plant Seedling dataset consists of 12 classes representing common plant species in Danish agriculture, and each of the 5,539 images in this dataset has different pixel sizes. The Asia Pacific Tele-Ophthalmology Society 2019 Blindness Detection (APTOS 2019 Blindness Detection) dataset consists of five classes with an imbalanced distribution to classify diabetic retinopathy (DR), and each of the 3,662 images in this dataset has different pixel sizes. Table 1 lists the image types and numbers in each class in the datasets.

Since the images in the Cifar-10 dataset have the same pixel dimensions and there is no common distinctiveness in the objects in the images, they were not subjected to any preprocessing. Various preprocesses were applied to the images in the V2 Plant Seedling dataset to ensure smoother progress of the training process. In the V2 Plant Seedling dataset images, plant greens can be distinguished in color space because they are denser than other objects. Therefore, the images in this dataset were masked by applying color-based segmentation. The best color range to distinguish plant greens in the color space was determined between RGB: 27-45-45 and RGB: 100-255-255. After masking, the images in the V2 Plant Seedling dataset were set to $224 \times 224 \times 3$ pixel dimensions. Since there is a class imbalance in the APTOS 2019 Blindness Detection dataset, no preprocessing other than dimension adjustment was performed to determine the behavior of the proposed model in an imbalanced dataset. Only the images in the dataset were set to

**Table 1 Image types and numbers in each class in the datasets.**

| Cifar-10 dataset image type | Number of image | V2 plant seedling dataset image type | Number of image | APTOS 2019 dataset image type | Number of image |
|---|---|---|---|---|---|
| Airplane | 6,000 | Black-grass | 309 | No DR | 1,805 |
| Automobile | 6,000 | Charlock | 452 | Mild DR | 370 |
| Bird | 6,000 | Cleavers | 335 | Moderate DR | 999 |
| Cat | 6,000 | Common Chickweed | 713 | Severe DR | 193 |
| Deer | 6,000 | Common wheat | 253 | Proliferate DR | 295 |
| Dog | 6,000 | Fat Hen | 538 | | |
| Frog | 6,000 | Loose Silky-bent | 762 | | |
| Horse | 6,000 | Maize | 257 | | |
| Ship | 6,000 | Scentless Mayweed | 607 | | |
| Truck | 6,000 | Shepherd's Purse | 274 | | |
| | | Small-flowered Cranesbill | 576 | | |
| | | Sugar beet | 463 | | |
| **Total** | **60,000** | **Total** | **5,539** | **Total** | **3,662** |

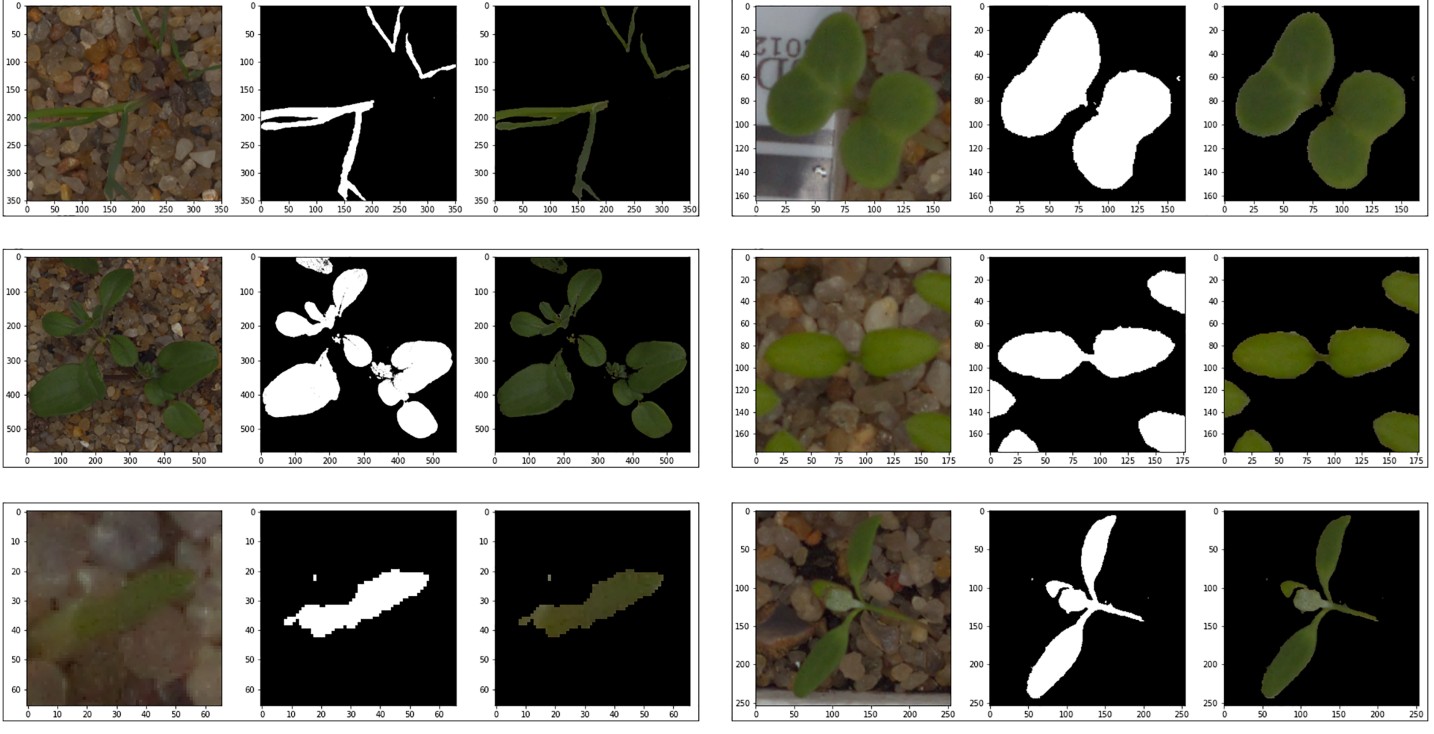

**Figure 1  Examples of masked images from the V2 plant seedling dataset.**

224 × 224 × 3 pixel dimensions. Figure 1 shows some examples obtained as a result of masking from images in the V2 Plant Seedling dataset, and Fig. 2 shows sample images of each class in the APTOS 2019 Blindness Detection dataset.

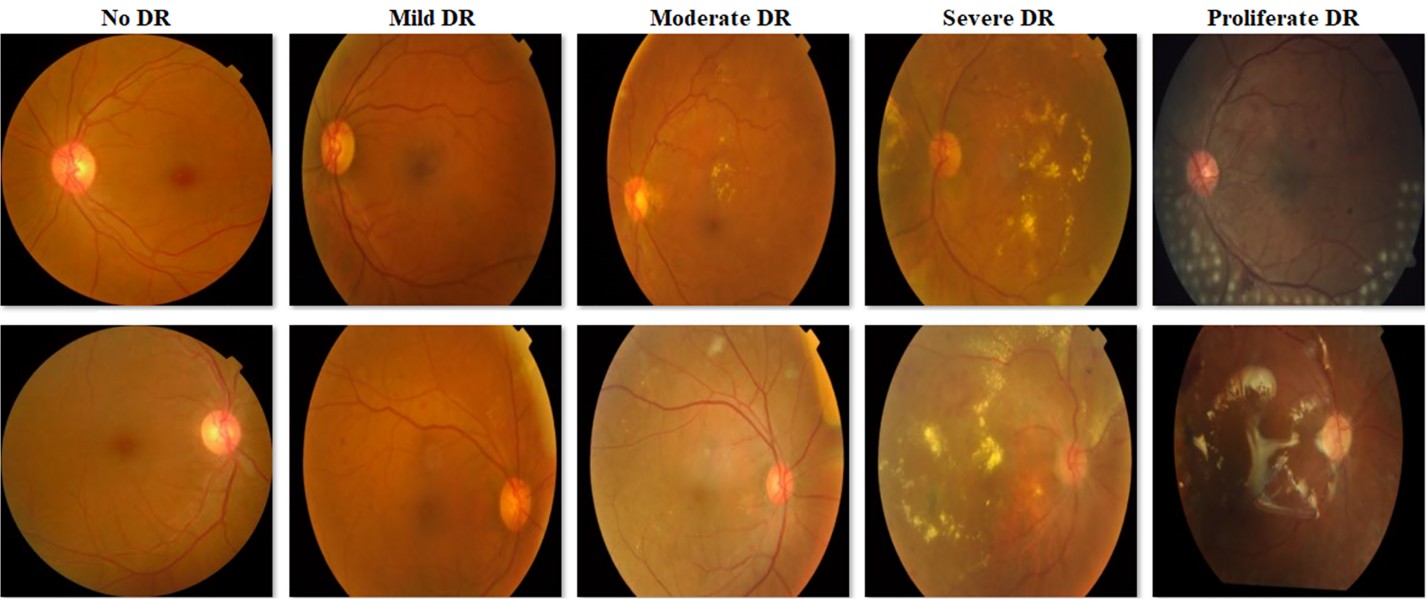

No DR    Mild DR    Moderate DR    Severe DR    Proliferate DR

**Figure 2** Examples of images from the APTOS 2019 Blindness Detection dataset.

## Model training parameters and performance metrics

In the experiments conducted in this study, the Adamax optimization algorithm was used at its default values. The loss function was implemented as Categorical Crossentropy and the classification activation function was implemented as Softmax. To prevent the overfitting tendency of the models during training, the minimum level of loss value in the validation dataset was checked with the "patience" feature used in the "early stopping" process. Therefore, a fixed epoch value was not used. The batch size value was set to 128 for training with the Cifar-10 dataset and 16 for training with the V2 Plant Seedling and APTOS 2019 Blindness Detection datasets. The Cifar-10 dataset is divided into two datasets 80% training, and 20% testing, and the V2 Plant Seedling and APTOS 2019 Blindness Detection datasets are divided into three datasets 70% training, 15% validation, and 15% testing.

After the model is created, various evaluation metrics are needed to measure how its performance works (*Sohan et al., 2019*). Evaluation metrics mostly come from the confusion matrix (*Zhang et al., 2016*). Therefore, accuracy, loss, precision, recall, and F1-score metrics were examined to determine the classification success of the models.

## AFCS-CNN model structure

AFCS-CNN model structure works on the principle of switching AF during model training. It suggests that AF can be switched cyclically depending on the performance decrease in the neural network. The workflow of the AFCS-CNN model structure is shown in Fig. 3, and the flowchart is shown in Algorithm 1.

The AFCS-CNN model structure, whose flow is shown in Fig. 3 and Algorithm 1, initially takes *list_AF*, $p_0$, $p_1$, *AFCS_loop* and *min_AF_count* parameter values as input. In the neural network (*ModelA*), the best and worst performing AFs are determined by trying

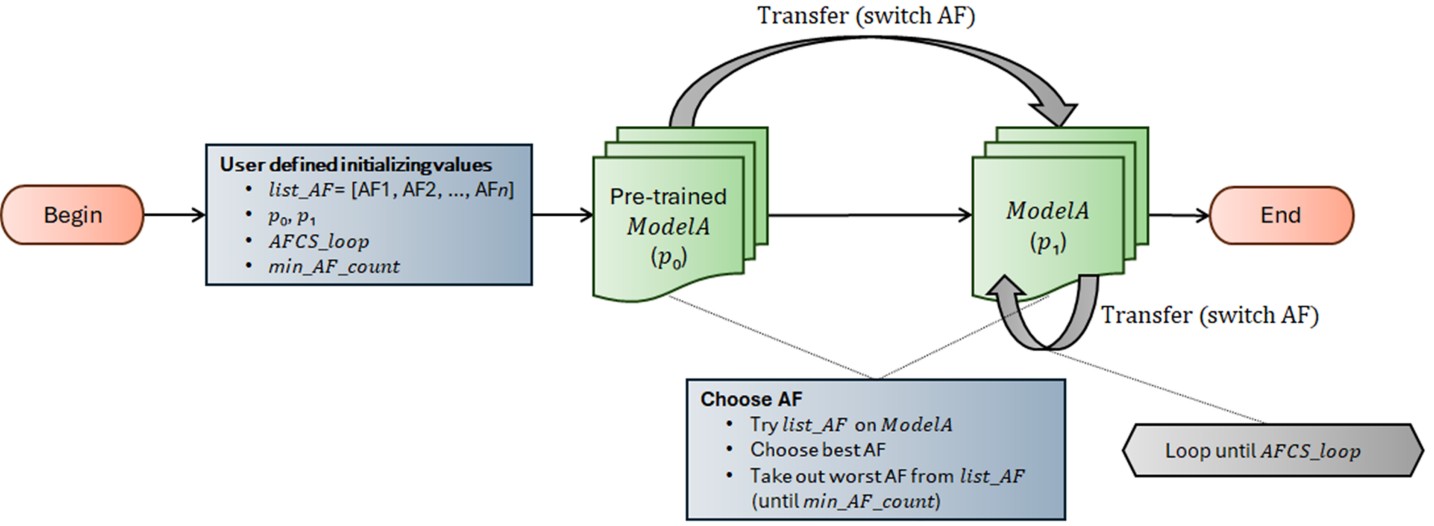

**Figure 3** Workflow of AFCS-CNN model structure.

---

**Algorithm 1** AFCS-CNN algorithm.

**begin**

  $list\_AF$ = [AF1, AF2, …, AF$n$]

  $p_0$, $p_1$, $AFCS\_loop \in [2, +\infty)$, $min\_AF\_count \in [2, +\infty)$

  try $list\_AF$ on $ModelA$ and choose best AF and take out worst AF from $list\_AF$

  start $ModelA$ training

  **if** patience = $p_0$ **then**

   stop $ModelA$ training

   **for** $i$ = 2 **to** $AFCS\_loop$ **step 1 do**

    try $list\_AF$ on $ModelA$ and choose next best AF

    **if** $min\_AF\_count$ < len ($list\_AF$) **then**

     take out worst AF from $list\_AF$

    **end if**

    continue $ModelA$ training with new AF

    **if** patience = $p_1$ **then**

     stop $ModelA$ training

    **end if**

   **end for**

  **end if**

**end**

---

them among a series of AFs ($list\_AF$) determined as parameters at the input. Training starts with the best-performing AF, while the worst-performing AF is taken out from $list\_AF$. During training, whether the performance of the model on the validation dataset

improves is checked with the patience ($p_0$, $p_1$) hyperparameter used in the early stopping process. In case the model performance does not improve during training, the best and the worst performing AFs are determined again among the remaining AFs. Training continues with the best-performing new AF. Take out process of the worst-performing AF from the system is continued cyclically until the minimum number of AFs (*min_AF_count*) defined by the user at the beginning is reached. Training is finished when the number of AFCS cycles (*AFCS_loop*) defined at the beginning is reached.

The runtime of the AFCS-CNN model structure, whose flow is shown in Fig. 3 and Algorithm 1, is given in Eq. (1).

$$Runtime_{AFCS-CNN} = Rt_{CNN} + Rt_{ep} \left( p_0 + \sum_{i=2}^{AFCS\_loop} p_1 \right) \tag{1}$$

where $Rt_{CNN}$ is the total runtime of any CNN model, $Rt_{ep}$ is an epoch runtime and $p_0$ and $p_1$ are patience values.

According to Eq. (1), the runtime of the AFCS-CNN model is calculated by adding the cost that the $p_0$ and $p_1$ values will bring to the total runtime of the CNN model to be used in the AFCS-CNN model. The additional cost is found by multiplying the sum of the $p_0$ value and the $p_1$ value (as many as the number of cycles) by an epoch runtime of the CNN model. Considering the cases where the patience hyperparameter is used during model training, the additional runtime cost of the AFCS-CNN model can be ignored if the $p_0$ and $p_1$ parameters are given small values. For example, let's assume that patience=10 is given in a CNN model that does not use the AFCS-CNN model structure and $p_0$=6, $p_1$=5 (*AFCS_loop*=5) in the AFCS-CNN model. In this case, while the CNN model will run 10 epochs more, the AFCS-CNN model will run 6+5*4=26 epochs more. But in the AFCS-CNN model, when $p_0$=3, $p_1$=2 (*AFCS_loop*=5) is given, the AFCS-CNN model will have run 3+2×4=11 epochs more. Or, when $p_0$=2, $p_1$=1 (*AFCS_loop*=5) is given in the AFCS-CNN model, the AFCS-CNN model will have run 2+1*4=6 epochs more. Therefore, it is recommended to give small values to the $p_0$ and $p_1$ parameters to obtain a competitive training cost with the AFCS-CNN model.

## RESULTS AND DISCUSSION

In this study, ablation studies were performed using the Cifar-10 dataset to determine the CNN models to be used in the proposed AFCS-CNN model structure and the special hyperparameters of the proposed model structure, and expansion experiments were performed using the V2 Plant Seedling and APTOS 2019 Blindness Detection datasets to determine the performance of the proposed model structure.

### Ablation studies

Ablation studies to determine the best CNN models and hyperparameters consist of three stages. The first stage includes determining the CNN models to be used in the AFCS-CNN model structure, the second stage includes determining the *list_AF* array and the $p_0$ parameter value, and the third stage includes determining the $p_1$ parameter value.

**Table 2 Numerical results obtained from model training in the Cifar-10 dataset.**

| Model | Default AF | Accuracy | Loss | Precision | Recall | F1-score |
|---|---|---|---|---|---|---|
| VGG16 | ReLU | 0.8452 | 0.8203 | 0.8460 | 0.8470 | 0.8450 |
| ResNet50 | ReLU | 0.8017 | 1.0741 | 0.8050 | 0.8030 | 0.8020 |
| MobileNet | Linear | 0.7928 | 1.0904 | 0.7918 | 0.7813 | 0.7855 |
| DenseNet121 | ReLU | 0.8293 | 0.9186 | 0.8350 | 0.8300 | 0.8290 |
| NASNetMobile | ReLU | 0.7499 | 1.4968 | 0.7520 | 0.7500 | 0.7480 |
| EfficientNetV2B0 | SiLU | 0.5294 | 2.6258 | 0.6320 | 0.5300 | 0.5190 |
| ConvNeXtTiny | GELU | 0.1000 | 2.3050 | 0.0100 | 0.1000 | 0.0180 |

### Stage 1: Determination of CNN models to be used in the AFCS-CNN model structure

To determine the CNN models to be used in the proposed AFCS-CNN model structure, VGG16 (*Simonyan & Zisserman, 2014*), ResNet50 (*He et al., 2016*), MobileNet (*Howard, 2017*), DenseNet121 (*Huang et al., 2017*), NASNetMobile (*Zoph et al., 2018*), EfficientNetV2B0 (*Tan & Le, 2021*), and ConvNeXtTiny (*Liu et al., 2022*) Keras models were tried on the Cifar-10 dataset. CNN models were trained for 20 epochs with pre-trained weights without partial layer freezing, and at default AF values. The numerical results obtained as a result of the training are given in Table 2.

According to the numerical results given in Table 2, the two best-performing models are VGG16 (84.52%) and DenseNet121 (82.93%), while the two worst-performing models are EfficientNetV2B0 (52.94%) and ConvNeXtTiny (10.00%). Since it would be appropriate to conduct ablation studies with the best-performing VGG16 and DenseNet121 models, these two models were preferred. It should be noted that in addition to these two best-performing models, the two worst-performing models (EfficientNetV2B0 and ConvNeXtTiny) were also used in the expansion experiments presented in the later sections of the study.

### Stage 2: Determination of list_AF array and $p_0$ parameter value

After determining the CNN models to be used in ablation studies, special hyperparameters of AFCS-CNN need to be defined. These hyperparameters can be given randomly or determined as a result of certain ablation studies. Here, a series of ablation studies were performed using the VGG16 and DenseNet121 models and the Cifar-10 dataset to determine the specific hyperparameters of AFCS-CNN.

Firstly, training was performed with VGG16 and DenseNet121 models to determine the *list_AF* array and the $p_0$ parameter value (epoch=20, patience=10). In addition to the ReLU, SiLU, and GELU AFs from the model default AF values presented in Table 2, the Tanh, ELU, SELU, and Mish AFs accepted in the literature were also preferred and the first version of the *list_AF* array was created. The VGG16 and DenseNet121 models use ReLU AF by default. The default AF of VGG16 and DenseNet121 models were trained separately by replacing them with the AFs in the *list_AF* array (fixed AF throughout training). The experiments were repeated three times to determine the best AFs. The test loss values

**Table 3 Test loss values obtained as a result of experiments to determine the best AFs.**

| Model | Exp | Epoch | AF | | | | | | |
|-------|-----|-------|------|------|------|------|------|------|------|
| | | | Tanh | ReLU | ELU | SELU | Mish | SiLU | GELU |
| VGG16 | Exp1 | 1 | 2.3076 | 0.9476 | 1.0678 | 1.2563 | 1.2892 | 1.1551 | 1.1570 |
| | | 2 | | 0.6696 | 0.6680 | 0.7820 | 0.8432 | 0.8062 | 0.7725 |
| | | 3 | | 0.5962 | 0.5724 | 0.6139 | 0.6929 | 0.6398 | 0.7045 |
| | | 4 | | 0.5256 | 0.5992 | 0.6085 | 0.6095 | 0.6129 | 0.6445 |
| | | 5 | | | 0.5696 | 0.6102 | 0.6128 | 0.5439 | 0.5598 |
| | | 6 | | | 0.5554 | 0.5163 | 0.5340 | | 0.5436 |
| | Exp2 | 1 | 2.3142 | 1.0733 | 0.9366 | 1.4203 | 1.2050 | 1.2574 | 1.3257 |
| | | 2 | 2.3071 | 0.6606 | 0.7246 | 0.8865 | 0.8317 | 0.7698 | 0.9488 |
| | | 3 | | 0.6901 | 0.5796 | 0.6664 | 0.5938 | 0.6802 | 0.6997 |
| | | 4 | | 0.5445 | 0.5255 | 0.5771 | 0.5877 | 0.5580 | 0.6504 |
| | | 5 | | 0.5327 | 0.6197 | 0.5706 | 0.5932 | 0.5556 | 0.5919 |
| | | 6 | | | 0.5106 | 0.5274 | 0.5679 | | 0.5578 |
| | Exp3 | 1 | 2.3181 | 1.3439 | 2.3096 | 1.2635 | 1.2560 | 1.2025 | 1.6888 |
| | | 2 | 2.3079 | 0.8153 | 2.3157 | 0.8118 | 0.7729 | 0.7723 | 0.8539 |
| | | 3 | | 0.7513 | 2.3073 | 0.6305 | 0.6168 | 0.6512 | 0.9304 |
| | | 4 | | 0.6177 | 1.3916 | 0.6437 | 0.5809 | 0.5713 | 0.6721 |
| | | 5 | | 0.5600 | 0.7189 | 0.5747 | 0.5858 | | 0.5833 |
| | | 6 | | 0.5151 | 0.6646 | | 0.5580 | | 0.6320 |
| | | 7 | | | 0.5391 | | | | 0.5170 |
| | | 8 | | | 0.5236 | | | | |
| DenseNet121 | Exp1 | 1 | 2.5334 | 0.6554 | 1.5114 | 1.1965 | 1.1738 | 1.1499 | 0.9080 |
| | | 2 | 1.3031 | 0.5978 | 1.1989 | 1.1150 | 1.0227 | 0.9996 | 0.6984 |
| | | 3 | 1.5998 | | 1.2462 | 1.0469 | 0.8134 | 0.9571 | 0.6139 |
| | | 4 | 1.1421 | | 0.8565 | 1.0753 | 1.1066 | 0.7643 | 0.6911 |
| | | 5 | 1.1080 | | 0.9678 | 0.9509 | 0.9248 | 1.0973 | 0.7137 |
| | | 6 | 1.0704 | | 0.8023 | 0.8480 | 0.6475 | 0.7673 | 0.5965 |
| | | 7 | 0.9837 | | 0.7692 | 0.8905 | | 0.7938 | |
| | | 8 | 0.8963 | | 0.8283 | 0.8551 | | 0.7441 | |
| | | 9 | 0.8924 | | 0.7902 | 0.8649 | | 0.8058 | |
| | | 10 | 1.2865 | | 0.7778 | 0.9231 | | 0.7149 | |
| | | 11 | 0.8584 | | 0.6953 | 0.8685 | | 0.6174 | |
| | | 12 | 20.381 | | | 0.9441 | | | |
| | | 13 | 0.9130 | | | 0.7886 | | | |
| | | 14 | 0.9475 | | | 0.8488 | | | |
| | | 15 | 1.5578 | | | 0.7172 | | | |
| | | 16 | 0.7754 | | | | | | |

| Table 3 (continued) | | | | | | | | | | |
| --- | --- | --- | --- | --- | --- | --- | --- | --- | --- | --- |
| Model | Exp | Epoch | AF | | | | | | | |
| | | | Tanh | ReLU | ELU | SELU | Mish | SiLU | GELU |
| | Exp2 | 1 | 2.6434 | 0.6864 | 1.2152 | 1.2862 | 1.3581 | 1.1517 | 0.8766 |
| | | 2 | 1.4907 | 0.6845 | 1.0475 | 1.0604 | 1.4940 | 1.1341 | 0.8051 |
| | | 3 | 1.8102 | 0.6161 | 0.8702 | 1.1703 | 0.7974 | 0.8132 | 0.6477 |
| | | 4 | 0.9409 | | 1.0633 | 1.1219 | 0.8704 | 0.8092 | 0.6355 |
| | | 5 | 1.4932 | | 0.9105 | 1.0914 | 0.7391 | 0.7236 | 0.8993 |
| | | 6 | 1.3614 | | 0.7997 | 0.9500 | 0.7348 | 0.7522 | 0.5716 |
| | | 7 | 0.9439 | | 0.8416 | 1.0224 | 0.6762 | 0.7218 | |
| | | 8 | 1.0116 | | 0.8869 | 1.2598 | 0.7274 | 0.7383 | |
| | | 9 | 0.9864 | | 0.9702 | 1.0666 | 0.7168 | 0.7116 | |
| | | 10 | 1.2859 | | 0.7945 | 1.2473 | 0.7997 | 0.6661 | |
| | | 11 | 0.7728 | | 0.7792 | 0.8475 | 0.6685 | 0.7161 | |
| | | 12 | | | 1.0680 | 0.7670 | | 0.6738 | |
| | | 13 | | | 0.7520 | | | 0.6274 | |
| | | 14 | | | 0.7260 | | | | |
| | Exp3 | 1 | 2.8821 | 0.6683 | 1.4735 | 1.2436 | 1.0554 | 1.0850 | 0.8528 |
| | | 2 | 1.2687 | 1.0291 | 0.9248 | 1.1282 | 1.1075 | 0.9101 | 0.9093 |
| | | 3 | 1.1558 | 0.5604 | 0.8414 | 1.0731 | 0.7743 | 0.8047 | 0.8000 |
| | | 4 | 1.1121 | | 0.8957 | 1.1002 | 0.8894 | 0.7906 | 0.7144 |
| | | 5 | 0.9807 | | 1.0537 | 0.9099 | 0.6880 | 0.7124 | 0.6606 |
| | | 6 | 0.9404 | | 0.7695 | 0.8878 | 0.6593 | 0.7021 | 0.6444 |
| | | 7 | 1.1520 | | 1.1104 | 0.8715 | 0.8545 | 0.7681 | 0.6788 |
| | | 8 | 0.9789 | | 0.8887 | 0.8372 | 0.5865 | 0.7868 | 0.6399 |
| | | 9 | 0.9592 | | 0.8042 | 0.9499 | | 0.6442 | |
| | | 10 | 0.8842 | | 0.7549 | 0.8418 | | | |
| | | 11 | 1.2925 | | 0.7156 | 1.1430 | | | |
| | | 12 | 0.7164 | | 0.8619 | 0.9685 | | | |
| | | 13 | | | 1.0142 | 0.9562 | | | |
| | | 14 | | | 0.7758 | 1.1994 | | | |
| | | 15 | | | 0.6857 | 0.8176 | | | |
| | | 16 | | | | 0.9022 | | | |
| | | 17 | | | | 0.9555 | | | |
| | | 18 | | | | 0.8060 | | | |

obtained as a result of experiments performed to measure the training performance of the models with each fixed AF are given in Table 3.

As a result of the experiments presented in Table 3, the Tanh function was removed from the AF list because learning did not occur with Tanh AF in the VGG16 model and the worst loss values were obtained with Tanh AF in the DenseNet121 model. Thus, the *list_AF* array was created as ReLU, ELU, SELU, Mish, SiLU, and GELU.

**Table 4 Consecutive loss increase numbers (determination of $p_0$ value).**

| Model | Experiment | AF | | | | | | Average |
|---|---|---|---|---|---|---|---|---|
| | | ReLU | ELU | SELU | Mish | SiLU | GELU | |
| VGG16 | Exp1 | 0 | 1 | 1 | 1 | 0 | 0 | 0.50 |
| | Exp2 | 1 | 1 | 0 | 1 | 0 | 0 | 0.50 |
| | Exp3 | 0 | 1 | 1 | 1 | 0 | 1 | 0.67 |
| | Overall average | | | | | | | 0.56 |
| DenseNet121 | Exp1 | 0 | 3 | 6 | 2 | 3 | 2 | 2.67 |
| | Exp2 | 0 | 3 | 4 | 3 | 2 | 1 | 2.17 |
| | Exp3 | 1 | 3 | 6 | 1 | 2 | 1 | 2.33 |
| | Overall average | | | | | | | 2.39 |

According to the experimental results given in Table 3, the consecutive loss increment numbers of the AFs were taken into account to determine the value of the $p_0$ parameter. By examining the loss increase situations in Table 3, the consecutive loss increase numbers were determined and summarized numerically in Table 4.

When Table 4 is examined, the overall average of consecutive loss increase numbers is calculated as 0.56 with the VGG16 model and 2.39 with the DenseNet121 model. Since training will be carried out with many models and datasets other than VGG16 and DenseNet121 models in the expansion experiments presented in the later sections of the study, the $p_0$ value was determined by taking into account the DenseNet121 model, which has the highest average of consecutive loss increase numbers. Since the consecutive loss increase number = 2.39≈2, the $p_0$ value is set to three. Thus, the loss will be allowed to increase two times in a row, and the training will be stopped at this stage when the loss value worsens for the 3rd time.

### Stage 3: Determination of the $p_1$ parameter value

After determining the *list_AF* array and $p_0$ parameter values, to determine the $p_1$ parameter value, the training was continued by selecting the three most successful AFs that reached the lowest loss value from the experiments performed with each model in Table 3 (epoch=20, patience=5). In this way, the behavior of the $p_1$ parameter values according to different AF antecedents was investigated. Among the experiments performed with VGG16, the three most successful AFs that brought the model to the lowest loss value were SELU, ReLU, Mish in experiment 1, ELU, SELU, ReLU in experiment 2, and ReLU, GELU, ELU in experiment 3. Among the experiments performed with DenseNet121, the three most successful AFs that brought the model to the lowest loss value were GELU, ReLU, SiLU in experiment 1 and experiment 2, and ReLU, Mish, GELU in experiment 3. Training was continued on the models obtained with these AFs, and experiments were carried out to determine the best AF in the next step. The test loss values obtained as a result of the experiments are given in Table 5.

According to the experimental results given in Table 5, the consecutive loss increment numbers of the AFs were taken into account to determine the value of the $p_1$ parameter. By

**Table 5 Test loss values obtained as a result of experiments to determine the next best AFs.**

| Model | Experiment | Previous best AFs | Epoch | Next AF | | | | | |
|---|---|---|---|---|---|---|---|---|---|
| | | | | ReLU | ELU | SELU | Mish | SiLU | GELU |
| VGG16 | Exp1 | SELU | 1 | 0.5176 | 0.5686 | | 0.5154 | 0.5488 | 0.5974 |
| | | | 2 | | 0.5863 | | | | |
| | | | 3 | | 0.5659 | | | | |
| | | ReLU | 1 | | 0.5669 | 0.8879 | 0.5360 | 0.5024 | 0.5180 |
| | | | 2 | | 0.5193 | 0.6527 | | | |
| | | | 3 | | | 0.6220 | | | |
| | | | 4 | | | 0.5375 | | | |
| | | Mish | 1 | 0.5685 | 0.6889 | 0.7069 | | 0.5997 | 0.5727 |
| | | | 2 | | 0.6410 | 0.5970 | | | |
| | | | 3 | | 0.5883 | | | | |
| | Exp2 | ELU | 1 | 0.5168 | | 0.5886 | 0.5549 | 0.5565 | 0.5648 |
| | | | 2 | | | | | 0.5424 | |
| | | SELU | 1 | 0.5955 | 0.5296 | | 0.6120 | 0.5394 | 0.5688 |
| | | | 2 | 0.5555 | | | 0.5665 | 0.5379 | 0.5490 |
| | | ReLU | 1 | | 0.7453 | 0.9466 | 0.5607 | 0.5521 | 0.5671 |
| | | | 2 | | 0.5835 | 0.6856 | | | |
| | | | 3 | | 0.5711 | 0.5991 | | | |
| | | | 4 | | 0.5508 | | | | |
| | Exp3 | ReLU | 1 | | 0.6359 | 0.6902 | 0.5166 | 0.5393 | 0.5562 |
| | | | 2 | | 0.5564 | 0.5660 | | 0.5208 | |
| | | GELU | 1 | 0.5698 | 0.8151 | 1.0327 | 0.5688 | 0.5495 | |
| | | | 2 | | 0.5771 | 0.6736 | 0.5681 | | |
| | | | 3 | | 0.5500 | 0.6185 | | | |
| | | | 4 | | 0.5921 | | | | |
| | | ELU | 1 | 0.5722 | | 0.5233 | 0.5318 | 0.5265 | 0.5046 |
| | | | 2 | 0.5171 | | | | | |
| DenseNet121 | Exp1 | GELU | 1 | 0.5478 | 0.9620 | 1.2328 | 0.9561 | 0.6372 | |
| | | | 2 | | 0.7391 | 1.1520 | 0.7870 | | |
| | | | 3 | | 0.8254 | 1.3142 | 0.6893 | | |
| | | | 4 | | 1.0896 | 1.0417 | 0.7205 | | |
| | | | 5 | | 0.7031 | 1.2832 | 0.6803 | | |
| | | | 6 | | | 1.2325 | | | |
| | | | 7 | | | 1.5440 | | | |
| | | | 8 | | | 0.8618 | | | |

(Continued)
| Table 5 (continued) | | | | | | | | | |
|---|---|---|---|---|---|---|---|---|---|
| Model | Experiment | Previous best AFs | Epoch | Next AF | | | | | |
| | | | | ReLU | ELU | SELU | Mish | SiLU | GELU |
| | | ReLU | 1 | | 1.5544 | 1.2104 | 0.9347 | 0.8831 | 0.8520 |
| | | | 2 | | 0.9110 | 1.0741 | 0.7624 | 0.7925 | 0.7353 |
| | | | 3 | | 0.9521 | 1.0381 | 0.7525 | 0.6902 | 0.6069 |
| | | | 4 | | 1.1389 | 1.2686 | 1.0301 | 0.6127 | 0.7015 |
| | | | 5 | | 0.9409 | 1.2387 | 0.8243 | 0.7614 | 0.6703 |
| | | | 6 | | 0.9092 | 0.9237 | 0.6854 | 0.7430 | 0.5824 |
| | | | 7 | | 0.8348 | 1.0289 | 0.7646 | 0.6335 | |
| | | | 8 | | 0.8329 | 0.8830 | 0.6800 | 0.6071 | |
| | | | 9 | | 0.7013 | 1.0159 | 0.6325 | 0.5949 | |
| | | | 10 | | 0.8792 | 0.7960 | | | |
| | | | 11 | | 0.8068 | | | | |
| | | | 12 | | 0.8731 | | | | |
| | | | 13 | | 0.7015 | | | | |
| | | | 14 | | 0.6589 | | | | |
| | | SiLU | 1 | 0.6180 | 1.1109 | 1.7046 | 0.7340 | | 0.6514 |
| | | | 2 | | 0.7586 | 1.2093 | 0.7088 | | |
| | | | 3 | | 0.6890 | 0.9131 | | | |
| | Exp2 | GELU | 1 | 0.5539 | 0.8230 | 1.3235 | 0.7502 | 0.6129 | |
| | | | 2 | | 1.0572 | 1.1054 | 0.6783 | | |
| | | | 3 | | 0.8220 | 1.0143 | 0.6193 | | |
| | | | 4 | | 0.8454 | 0.9445 | | | |
| | | | 5 | | 0.7590 | 0.8853 | | | |
| | | | 6 | | 0.7906 | 0.9024 | | | |
| | | | 7 | | 0.7254 | 0.9238 | | | |
| | | | 8 | | | 1.1919 | | | |
| | | | 9 | | | 1.1035 | | | |
| | | | 10 | | | 0.8324 | | | |
| | | ReLU | 1 | | 1.0134 | 1.2213 | 0.9434 | 0.8702 | 0.9008 |
| | | | 2 | | 0.9379 | 1.1288 | 0.7861 | 0.8505 | 0.7791 |
| | | | 3 | | 1.0422 | 0.9406 | 0.6726 | 0.9180 | 0.6313 |
| | | | 4 | | 0.9784 | 1.0536 | 0.9586 | 0.6715 | 0.6604 |
| | | | 5 | | 0.8998 | 1.1468 | 0.7218 | 0.5984 | 0.7107 |
| | | | 6 | | 0.8033 | 0.9460 | 0.7172 | | 0.6131 |
| | | | 7 | | 0.8312 | 0.8926 | 0.8329 | | 0.6058 |
| | | | 8 | | 0.7211 | 0.9610 | 0.6197 | | 0.5496 |
| | | | 9 | | | 0.8678 | | | |
| Table 5 (continued) | | | | | | | | | |
|---|---|---|---|---|---|---|---|---|---|
| **Model** | **Experiment** | **Previous best AFs** | **Epoch** | **Next AF** | | | | | |
| | | | | **ReLU** | **ELU** | **SELU** | **Mish** | **SiLU** | **GELU** |
| | | SiLU | 1 | 0.6232 | 0.9390 | 1.5974 | 0.7148 | | 0.6366 |
| | | | 2 | | 0.8063 | 1.1322 | | | |
| | | | 3 | | 1.0008 | 0.9104 | | | |
| | | | 4 | | 0.7552 | 0.9664 | | | |
| | | | 5 | | 0.8774 | 1.0150 | | | |
| | | | 6 | | 0.8153 | 1.1679 | | | |
| | | | 7 | | 0.7441 | 0.8454 | | | |
| | Exp3 | ReLU | 1 | | 1.1103 | 1.1455 | 0.9927 | 1.1415 | 0.9124 |
| | | | 2 | | 1.3042 | 1.1690 | 0.8607 | 0.8905 | 0.6170 |
| | | | 3 | | 0.8980 | 1.1342 | 0.7389 | 0.6987 | 0.7598 |
| | | | 4 | | 0.8322 | 0.9823 | 0.6869 | 0.6999 | 0.6880 |
| | | | 5 | | 0.8283 | 0.9458 | 0.6260 | 0.7668 | 0.5713 |
| | | | 6 | | 1.0211 | 1.3660 | 0.6776 | 0.5890 | |
| | | | 7 | | 0.8523 | 0.9618 | 0.6410 | | |
| | | | 8 | | 0.8341 | 1.2932 | 0.7156 | | |
| | | | 9 | | 0.8244 | 0.9192 | 0.8070 | | |
| | | | 10 | | 0.8018 | 0.7769 | 0.6258 | | |
| | | | 11 | | 0.6888 | | | | |
| | | | 12 | | 0.7590 | | | | |
| | | | 13 | | 0.6685 | | | | |
| | | Mish | 1 | 0.5551 | 0.8781 | 0.9893 | | 0.6514 | 0.6946 |
| | | | 2 | | 0.9072 | 0.9175 | | | 0.6533 |
| | | | 3 | | 0.8157 | | | | |
| | | | 4 | | 0.7380 | | | | |
| | | GELU | 1 | 0.5971 | 0.8531 | 1.3561 | 0.6342 | 0.9523 | |
| | | | 2 | | 0.8778 | 1.0365 | | 0.7361 | |
| | | | 3 | | 0.9276 | | | 0.7168 | |
| | | | 4 | | 0.8790 | | | 0.7327 | |
| | | | 5 | | 0.9445 | | | 0.7413 | |
| | | | 6 | | 0.7341 | | | 0.6974 | |

examining the loss increase situations in Table 5, the consecutive loss increase numbers were determined and summarized numerically in Table 6.

When Table 6 is examined, the overall average of consecutive loss increase numbers is calculated as 0.02 with the VGG16 model and 1.38 with the DenseNet121 model. The $p_1$ value was determined by taking into account the DenseNet121 model, which has the highest average of consecutive loss increase numbers. Since the consecutive loss increase number = $1.38 \approx 1$, the $p_1$ value was set to two. Thus, the loss will be allowed to increase one

**Table 6 Consecutive loss increase numbers (determination of $p_1$ value).**

| Model | Experiment | Previous best AFs | Next AF | | | | | | Average |
|---|---|---|---|---|---|---|---|---|---|
| | | | ReLU | ELU | SELU | Mish | SiLU | GELU | |
| VGG16 | Exp1 | SELU | 0 | 1 | | 0 | 0 | 0 | 0.20 |
| | | ReLU | | 0 | 0 | 0 | 0 | 0 | 0.00 |
| | | Mish | 0 | 0 | 0 | | 0 | 0 | 0.00 |
| | Exp2 | ELU | 0 | | 0 | 0 | 0 | 0 | 0.00 |
| | | SELU | 0 | 0 | | 0 | 0 | 0 | 0.00 |
| | | ReLU | | 0 | 0 | 0 | 0 | 0 | 0.00 |
| | Exp3 | ReLU | | 0 | 0 | 0 | 0 | 0 | 0.00 |
| | | GELU | 0 | 0 | 0 | 0 | 0 | | 0.00 |
| | | ELU | 0 | | 0 | 0 | 0 | 0 | 0.00 |
| | | Overall average | | | | | | | 0.02 |
| DenseNet121 | Exp1 | GELU | 0 | 2 | 3 | 1 | 0 | | 1.20 |
| | | ReLU | | 4 | 2 | 2 | 3 | 2 | 2.60 |
| | | SiLU | 0 | 0 | 0 | 0 | | 0 | 0.00 |
| | Exp2 | GELU | 0 | 1 | 4 | 0 | 0 | | 1.00 |
| | | ReLU | | 2 | 3 | 4 | 1 | 2 | 2.40 |
| | | SiLU | 0 | 2 | 3 | 0 | | 0 | 1.00 |
| | Exp3 | ReLU | | 3 | 3 | 4 | 2 | 2 | 2.80 |
| | | Mish | 0 | 1 | 0 | | 0 | 0 | 0.20 |
| | | GELU | 0 | 4 | 0 | 0 | 2 | | 1.20 |
| | | Overall average | | | | | | | 1.38 |

time in a row, and the training will be stopped at this stage when the loss value worsens for the 2nd time.

In summary, as a result of ablation studies, the *list_AF* array was created by taking the ReLU, ELU, SELU, Mish, SiLU, and GELU AFs where learning occurred. Thus, the initial values of the hyperparameter *list_AF*, which holds the AF list to be applied to the CNN model to be used in the AFCS-CNN model structure were defined. The values of the patience hyperparameters used in the early stopping process were determined as three at the beginning and two in the next cycles. Patience values were kept small to prevent early overfitting of the model. Thus, the best patience values are defined as $p_0=3$, $p_1=2$. When the DenseNet121 model is used in the AFCS-CNN model structure, it is defined as *AFCS_loop*=5 because the learning effect decreases after a certain cycle. The value of *min_AF_count* was determined as 3, allowing the opportunity to select from at least three AFs during switches.

## Expansion experiments

After ablation studies, VGG16, VGG19, DenseNet121, DenseNet169, EfficientNetV2B0, EfficientNetV2B1, ConvNeXtTiny, and ConvNeXtSmall CNN models were used in the proposed AFCS-CNN model structure to determine the performance success of the AFCS-

**Table 7 Success accuracy and loss values on the V2 plant seedling validation dataset.**

| Model | Metric | AF | | | | | | AFCS-CNN |
|---|---|---|---|---|---|---|---|---|
| | | ReLU | ELU | SELU | Mish | SiLU | GELU | |
| VGG16 | Accuracy (%) | 92.30 | 12.88 | 12.88 | 92.90 | 91.46 | 92.42 | 93.98 |
| | Loss | 0.2619 | 2.4264 | 2.4710 | 0.2255 | 0.2894 | 0.3810 | 0.1839 |
| VGG19 | Accuracy (%) | 87.48 | 13.72 | 88.33 | 89.89 | 90.85 | 92.06 | 93.50 |
| | Loss | 0.3778 | 2.4549 | 0.5133 | 0.3912 | 0.3053 | 0.2999 | 0.2084 |
| DenseNet121 | Accuracy (%) | 96.39 | 89.65 | 77.26 | 93.02 | 91.34 | 96.27 | 96.63 |
| | Loss | 0.1526 | 0.3514 | 0.7537 | 0.2808 | 0.3059 | 0.1605 | 0.1149 |
| DenseNet169 | Accuracy (%) | 96.03 | 80.02 | 68.83 | 92.78 | 90.25 | 93.62 | 96.39 |
| | Loss | 0.1559 | 0.5913 | 1.5005 | 0.2246 | 0.3579 | 0.2236 | 0.1296 |
| EfficientNetV2B0 | Accuracy (%) | 94.58 | 91.58 | 70.04 | 87.24 | 73.16 | 64.74 | 95.07 |
| | Loss | 0.2692 | 0.3450 | 1.1031 | 0.4215 | 1.0297 | 1.4183 | 0.2192 |
| EfficientNetV2B1 | Accuracy (%) | 93.86 | 71.72 | 74.01 | 91.34 | 91.70 | 82.79 | 95.07 |
| | Loss | 0.3060 | 0.9812 | 0.9816 | 0.3751 | 0.3325 | 0.5350 | 0.2225 |
| ConvNeXtTiny | Accuracy (%) | No learning occurred in any of the trainings | | | | | | |
| | Loss | | | | | | | |
| ConvNeXtSmall | Accuracy (%) | No learning occurred in any of the trainings | | | | | | |
| | Loss | | | | | | | |

CNN model structure. The hyperparameter values of the AFCS-CNN model structure are defined as determined in the previous section. So $list\_AF$ = [ReLU, ELU, SELU, Mish, SiLU, GELU], $p_0$=3, $p_1$=2, $AFCS\_loop$=5, $min\_AF\_count$=3 was set, and expansion experiments were performed on the V2 Plant Seedling and APTOS 2019 Blindness Detection datasets using VGG16, VGG19, DenseNet121, DenseNet169, EfficientNetV2B0, EfficientNetV2B1, ConvNeXtTiny, and ConvNeXtSmall CNN models in the AFCS-CNN model structure.

### Expansion experiments with plant seedling dataset

To determine the performance success of the AFCS-CNN model structure on the V2 Plant Seedling dataset, experiments were performed using VGG16, VGG19, DenseNet121, DenseNet169, EfficientNetV2B0, EfficientNetV2B1, ConvNeXtTiny, and ConvNeXtSmall CNN models, respectively, in the AFCS-CNN model structure. In this way, the effect of AF switching during model training was examined. To compare the results, the default AF of each model was replaced by each AF in the $list\_AF$ array, and training was performed separately for each case (using the same AF throughout training). Experiments were performed using the train-validation dataset for the AFCS-CNN model structure and each fixed AF and the classification accuracy and loss values of the models in the V2 Plant Seedling validation dataset are given in Table 7.

When the validation accuracy and loss values given in Table 7 are examined, it is seen that using all CNN models discussed in the study in the AFCS-CNN model structure gives more successful results than not using them. The accuracy rates and loss values in the

training performed using fixed AFs with CNN models without using the AFCS-CNN model structure were lower. Thanks to the AFCS-CNN model structure, it has been observed that switching AF provides a performance increase compared to fixing. While the most successful classification accuracy was achieved using the DenseNet121 model in the AFCS-CNN model (96.63%), no learning occurred in any of the training performed with ConvNeXt models. According to the validation success accuracies presented in Table 7, the accuracy and loss graphs of the three CNN models with the best performing AF and each CNN model used in the AFCS-CNN model structure are shown comparatively in Fig. 4.

In Fig. 4, graphs of the training performed with the most successful AFs that were left fixed throughout the training of each CNN model and the training performed using each CNN model in the AFCS-CNN model structure are shown together. As can be seen in the figure, the proposed AFCS-CNN model structure performs the training process in an up-and-down structure. The ups and downs in training the AFCS-CNN model structure are due to the switching of AF. For example, in the case where the VGG16 model was used in the AFCS-CNN model structure, the AFCS-CNN model structure started training with GELU AF, which performs best among the values in the *list_AF* array. Since the validation loss value increased three times ($p_0$ control) in a row after the 13th epoch, AF=Mish was made in this epoch and the training was continued from the 13th epoch. Since the worst performing AF was SELU, this AF was taken out from the *list_AF* array to not be used in the next switch. Again, after the 15th epoch, when the validation loss value increased two times ($p_1$ control), AF=SiLU was made, and the training was continued from the 15th epoch. At this stage, ELU AF was taken out from the *list_AF* array. Then, in the 17th epoch, AF=Mish was made, and ReLU AF was taken out from the *list_AF* array. In the 20th epoch, AF=GELU was made, and the cycle was finished in the 24th epoch. As seen in Fig. 4, in the training performed using only the VGG16 model and fixed AFs without using the AFCS-CNN model structure, it was observed that the difference between the train-validation graph widened after a certain epoch and overfitting occurred. It was determined that when the VGG16 model is used in the AFCS-CNN model structure, overfitting is prevented, and learning occurs in fewer epochs. Thus, in the training performed with the VGG16 model, it was observed that the AF switch during the training process with the AFCS-CNN model structure had a positive effect on the model training process.

In addition, it was observed that when other CNN models discussed in the study were used in the AFCS-CNN model structure, different AF switches occurred at different epochs and the AF switch had a positive effect on the model training process. When the train-validation accuracy and loss graphs shown in Fig. 4 are examined, the models made five AF switches depending on the number of cycles defined at the beginning. In the training of the CNN models used in the AFCS-CNN model structure on the V2 Plant Seedling dataset, different AF switches were made at different epochs, and the details are summarized in Table 8.

After the train-validation process, the final performance of the models was tested using the V2 Plant Seedling test dataset, which the network had never seen, and the numerical results are given in Table 9. Additionally, Fig. 5 shows the confusion matrix results

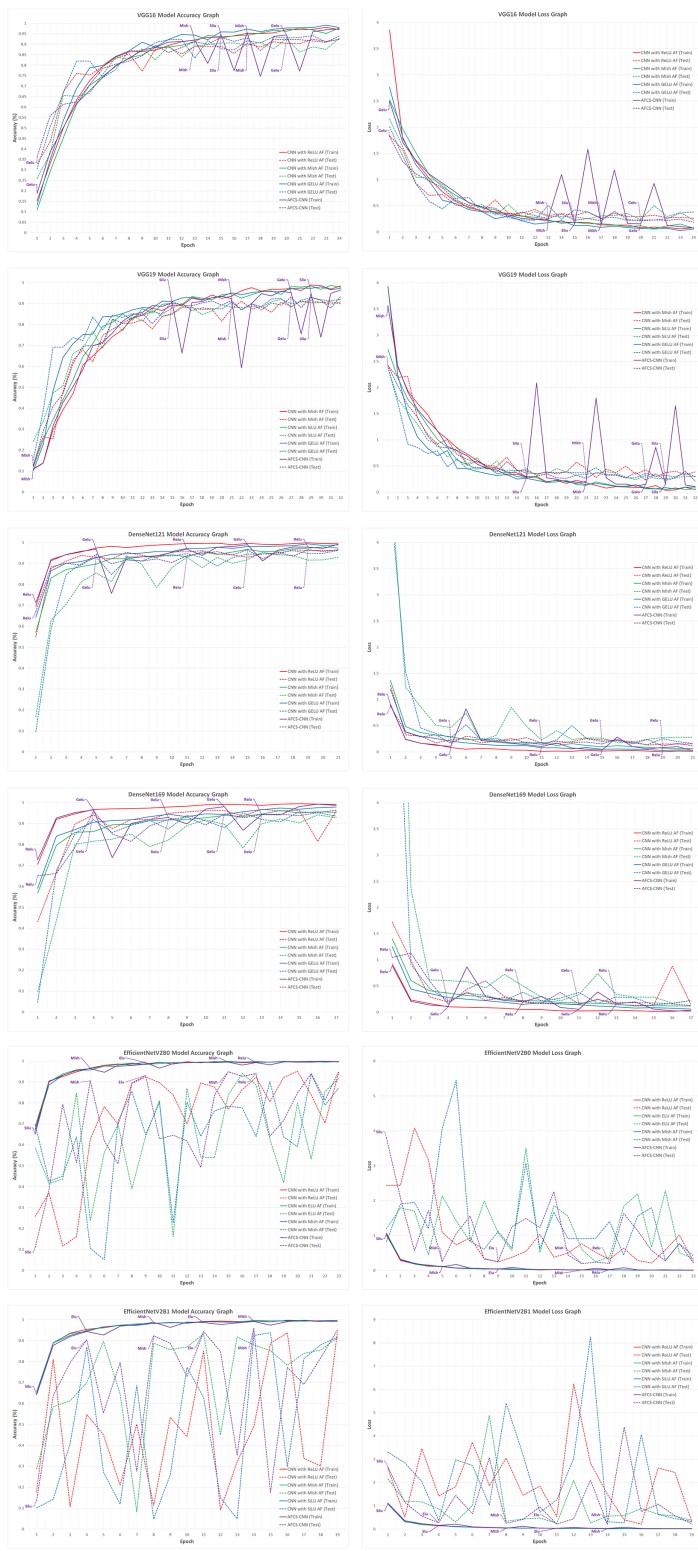

**Figure 4  Train-validation accuracy and loss graphs of each CNN model on the V2 plant seedling validation dataset.**               

**Table 8  AF switches in the training of the CNN models used in the AFCS-CNN model structure on the V2 plant seedling dataset.**

| AFCS-CNN (Model) | Step1 | | Step2 | | Step3 | | Step4 | | Step5 | |
|---|---|---|---|---|---|---|---|---|---|---|
| | Epoch | AF | Epoch | AF | Epoch | AF | Epoch | AF | Epoch | AF |
| AFCS-CNN (VGG16) | 1 | GELU | 13 | Mish | 15 | SiLU | 17 | Mish | 20 | GELU |
| AFCS-CNN (VGG19) | 1 | Mish | 15 | SiLU | 21 | Mish | 27 | GELU | 29 | SiLU |
| AFCS-CNN (DenseNet121) | 1 | ReLU | 5 | GELU | 11 | ReLU | 15 | GELU | 19 | ReLU |
| AFCS-CNN (DenseNet169) | 1 | ReLU | 4 | GELU | 8 | ReLU | 11 | GELU | 13 | ReLU |
| AFCS-CNN (EfficientNetV2B0) | 1 | SiLU | 5 | Mish | 9 | ELU | 15 | Mish | 17 | ReLU |
| AFCS-CNN (EfficientNetV2B1) | 1 | SiLU | 4 | ELU | 8 | Mish | 11 | ELU | 14 | Mish |

**Table 9  Numerical results obtained from models using the V2 plant seedling test dataset.**

| Model | Metric | AF | | | | | | AFCS-CNN |
|---|---|---|---|---|---|---|---|---|
| | | ReLU | ELU | SELU | Mish | SiLU | GELU | |
| VGG16 | Accuracy | 0.9193 | 0.1287 | 0.1287 | 0.9193 | 0.9133 | 0.9157 | 0.9362 |
| | Loss | 0.2923 | 2.4259 | 2.4699 | 0.2602 | 0.2903 | 0.3782 | 0.2074 |
| | Precision | 0.9215 | 0.0167 | 0.0167 | 0.9213 | 0.9155 | 0.9201 | 0.9354 |
| | Recall | 0.9195 | 0.1288 | 0.1288 | 0.9194 | 0.9141 | 0.9154 | 0.9367 |
| | F1-score | 0.9205 | 0.0296 | 0.0296 | 0.9202 | 0.9090 | 0.9104 | 0.9368 |
| VGG19 | Accuracy | 0.8916 | 0.1371 | 0.8676 | 0.9025 | 0.8977 | 0.9109 | 0.9253 |
| | Loss | 0.5742 | 2.4543 | 0.5411 | 0.3799 | 0.3579 | 0.2915 | 0.2368 |
| | Precision | 0.8934 | 0.0192 | 0.8652 | 0.9061 | 0.8987 | 0.9112 | 0.9259 |
| | Recall | 0.8917 | 0.1372 | 0.8668 | 0.9010 | 0.8989 | 0.9112 | 0.9256 |
| | F1-score | 0.8824 | 0.0329 | 0.8596 | 0.9030 | 0.8961 | 0.9096 | 0.9224 |
| DenseNet121 | Accuracy | 0.9542 | 0.8868 | 0.7496 | 0.9386 | 0.9157 | 0.9590 | 0.9711 |
| | Loss | 0.1581 | 0.3422 | 0.7506 | 0.2014 | 0.2752 | 0.1367 | 0.0904 |
| | Precision | 0.9555 | 0.8919 | 0.8280 | 0.9431 | 0.9503 | 0.9598 | 0.9719 |
| | Recall | 0.9552 | 0.8881 | 0.7492 | 0.9385 | 0.9160 | 0.9590 | 0.9714 |
| | F1-score | 0.9539 | 0.8817 | 0.7423 | 0.9397 | 0.9186 | 0.9574 | 0.9721 |
| DenseNet169 | Accuracy | 0.9446 | 0.7785 | 0.6594 | 0.9049 | 0.8880 | 0.9410 | 0.9675 |
| | Loss | 0.1971 | 0.6784 | 1.5014 | 0.2827 | 0.3053 | 0.1905 | 0.0955 |
| | Precision | 0.9437 | 0.8260 | 0.7670 | 0.9117 | 0.8981 | 0.9465 | 0.9682 |
| | Recall | 0.9454 | 0.7784 | 0.6599 | 0.9063 | 0.8883 | 0.9421 | 0.9688 |
| | F1-score | 0.9439 | 0.7701 | 0.6190 | 0.9044 | 0.8815 | 0.9416 | 0.9670 |
| EfficientNetV2B0 | Accuracy | 0.9265 | 0.9217 | 0.6943 | 0.8543 | 0.7075 | 0.6401 | 0.9518 |
| | Loss | 0.3443 | 0.3035 | 1.1257 | 0.5348 | 1.0339 | 1.3997 | 0.1822 |
| | Precision | 0.9334 | 0.9254 | 0.7661 | 0.8815 | 0.7790 | 0.7297 | 0.9508 |
| | Recall | 0.9271 | 0.9223 | 0.6951 | 0.8550 | 0.7067 | 0.6410 | 0.9527 |
| | F1-score | 0.9264 | 0.9236 | 0.6833 | 0.8515 | 0.7007 | 0.6199 | 0.9517 |

| Table 9 (continued) | | | | | | | | |
|---|---|---|---|---|---|---|---|---|
| Model | Metric | AF | | | | | | AFCS-CNN |
| | | ReLU | ELU | SELU | Mish | SiLU | GELU | |
| EfficientNetV2B1 | Accuracy | 0.9265 | 0.7148 | 0.7352 | 0.9169 | 0.9121 | 0.8122 | 0.9530 |
| | Loss | 0.3025 | 0.9967 | 0.9698 | 0.3140 | 0.3973 | 0.6153 | 0.1796 |
| | Precision | 0.9273 | 0.7581 | 0.8119 | 0.9224 | 0.9258 | 0.8288 | 0.9531 |
| | Recall | 0.9278 | 0.7150 | 0.7360 | 0.9167 | 0.9124 | 0.8124 | 0.9526 |
| | F1-score | 0.9241 | 0.7041 | 0.7274 | 0.9192 | 0.9145 | 0.7987 | 0.9518 |

obtained from the test dataset with each CNN model used in the AFCS-CNN model structure.

According to the results given in Table 9 and Fig. 5, the most successful classification accuracy in the V2 Plant Seedling test dataset was achieved by using the DenseNet121 model in the AFCS-CNN model. With the AFCS-CNN (DenseNet121) model, 97.11% accuracy and 0.0904 loss values were obtained in the test dataset. It has been observed that using all CNN models discussed in the study in the AFCS-CNN model structure gives more successful results than using fixed AF in CNN models. Thus, it has been demonstrated that the AFCS-CNN model structure provides an important performance increase in all CNN models discussed in the study.

### Expansion experiments with APTOS 2019 Blindness Detection dataset

To determine the performance success of the AFCS-CNN model structure on the APTOS 2019 Blindness Detection dataset, experiments were performed using VGG16, VGG19, DenseNet121, DenseNet169, EfficientNetV2B0, EfficientNetV2B1, ConvNeXtTiny, and ConvNeXtSmall CNN models, respectively, in the AFCS-CNN model structure. In this way, the effect of AF switching during model training was examined using a different dataset. To compare the results, the default AF of each model was replaced by each AF in the *list_AF* array, and training was performed separately for each case (using the same AF throughout training). Experiments were performed using the train-validation dataset for the AFCS-CNN model structure and each fixed AF and the classification accuracy and loss values of the models in the APTOS 2019 Blindness Detection validation dataset are given in Table 10.

When the validation accuracy and loss values given in Table 10 are examined, it is seen that using all CNN models, except VGG models, in the AFCS-CNN model structure gives more successful results than not using them. The accuracy rates and loss values in the training performed using fixed AFs with CNN models without using the AFCS-CNN model structure were lower. Thanks to the AFCS-CNN model structure, it has been observed that switching AF provides a performance increase compared to fixing. While the most successful classification accuracy was achieved using the DenseNet169 model in the AFCS-CNN model (83.45%), no learning occurred in any of the training performed with ConvNeXt models. According to the validation success accuracies presented in Table 10,

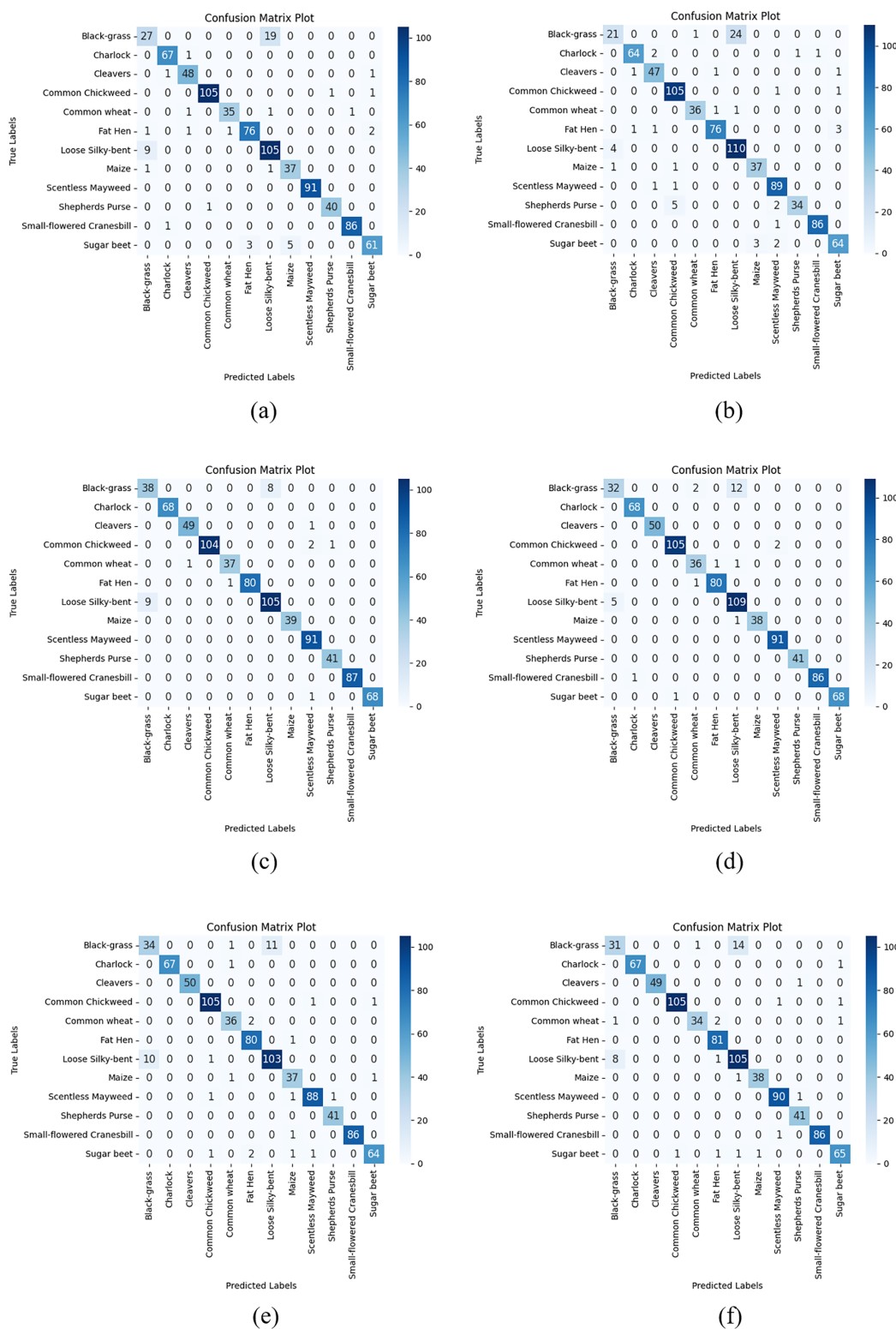

**Figure 5  Confusion matrix plots of each CNN model used in the AFCS-CNN model structure on the V2 plant seedling test dataset: (A) AFCS-CNN (VGG16), (B) AFCS-CNN (VGG19), (C) AFCS-CNN (DenseNet121), (D) AFCS-CNN (DenseNet169), (E) AFCS-CNN (EfficientNetV2B0), (F) AFCS-CNN (EfficientNetV2B1).**

**Table 10 Success accuracy and loss values on the APTOS 2019 Blindness Detection Validation dataset.**

| Model | Metric | AF | | | | | | AFCS-CNN |
|---|---|---|---|---|---|---|---|---|
| | | ReLU | ELU | SELU | Mish | SiLU | GELU | |
| VGG16 | Accuracy (%) | 76.91 | 69.09 | 75.27 | 80.73 | 77.82 | 77.09 | 80.18 |
| | Loss | 1.1004 | 0.8905 | 0.6830 | 0.6732 | 0.6521 | 1.0028 | 0.5188 |
| VGG19 | Accuracy (%) | 78.73 | 74.91 | 75.45 | 76.18 | 73.82 | 78.18 | 78.73 |
| | Loss | 0.6350 | 0.7446 | 0.6775 | 0.6536 | 0.7559 | 0.5691 | 0.5652 |
| DenseNet121 | Accuracy (%) | 81.82 | 74.00 | 76.36 | 77.45 | 72.91 | 78.55 | 83.27 |
| | Loss | 0.8153 | 0.7373 | 0.6211 | 0.6366 | 0.9755 | 0.7211 | 0.5611 |
| DenseNet169 | Accuracy (%) | 79.09 | 64.36 | 57.64 | 69.45 | 77.09 | 80.73 | 83.45 |
| | Loss | 0.9562 | 1.0919 | 1.3097 | 0.9848 | 0.6766 | 0.7533 | 0.6115 |
| EfficientNetV2B0 | Accuracy (%) | 62.55 | 74.18 | 59.27 | 74.91 | 70.91 | 70.91 | 79.45 |
| | Loss | 1.1221 | 0.9009 | 1.2356 | 0.9989 | 1.5562 | 1.0489 | 0.8403 |
| EfficientNetV2B1 | Accuracy (%) | 69.27 | 62.55 | 66.91 | 79.64 | 73.27 | 73.45 | 81.82 |
| | Loss | 1.0717 | 1.2512 | 1.0693 | 1.1388 | 1.0198 | 1.0289 | 0.8520 |
| ConvNeXtTiny | Accuracy (%) | No learning occurred in any of the trainings | | | | | | |
| | Loss | | | | | | | |
| ConvNeXtSmall | Accuracy (%) | No learning occurred in any of the trainings | | | | | | |
| | Loss | | | | | | | |

the accuracy and loss graphs of the three CNN models with the best performing AF and each CNN model used in the AFCS-CNN model structure are shown comparatively in Fig. 6.

In Fig. 6, graphs of the training performed with the most successful AFs that were left fixed throughout the training of each CNN model and the training performed using each CNN model in the AFCS-CNN model structure are shown together. As can be seen in the figure, similar to the previous analyses, the proposed AFCS-CNN model structure performs the training process in an up-and-down structure. The ups and downs in training the AFCS-CNN model structure are due to the switching of AF. For example, in the case where the VGG16 model was used in the AFCS-CNN model structure, the AFCS-CNN model structure started training with ReLU AF, which performs best among the values in the *list_AF* array. Since the validation loss value increased three times ($p_0$ control) in a row after the 11th epoch, AF=SiLU was made in this epoch and the training was continued from the 11th epoch. Since the worst performing AF was ELU, this AF was taken out from the *list_AF* array to not be used in the next switch. Again, after the 16th epoch, when the validation loss value increased two times ($p_1$ control), AF=ReLU was made, and the training was continued from the 16th epoch. At this stage, SELU AF was taken out from the *list_AF* array. Then, in the 28th epoch, AF=Mish was made, and GELU AF was taken out from the *list_AF* array. In the 32nd epoch, AF=SiLU was made, and the cycle was finished in the 35th epoch. As seen in Fig. 6, in the training performed using only the VGG16 model and fixed AFs without using the AFCS-CNN model structure, it was observed that the difference between the train-validation graph widened after a certain

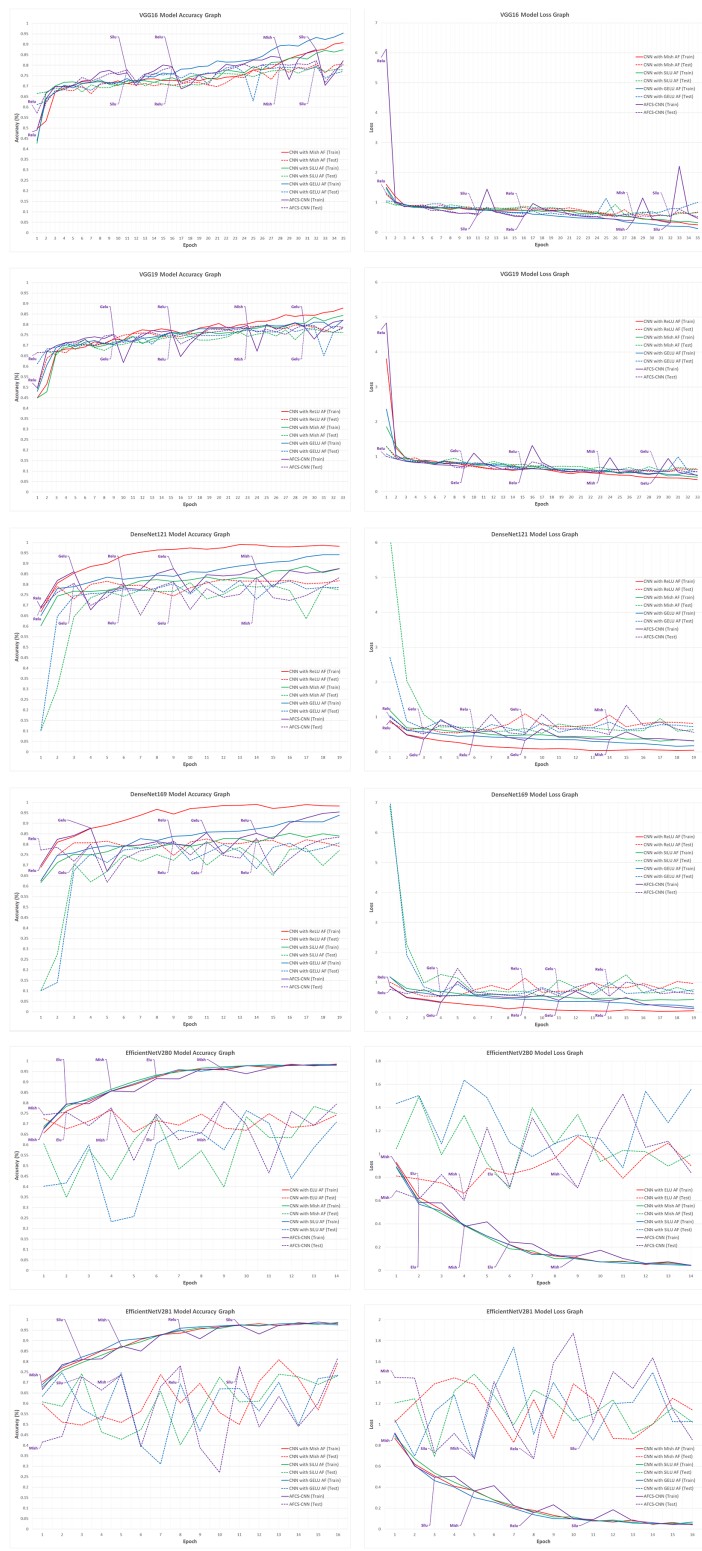

**Figure 6 Train-validation accuracy and loss graphs of each CNN model on the APTOS 2019 Blindness Detection dataset.**

**Table 11  AF switches in the training of CNN models used in the AFCS-CNN model structure on the APTOS 2019 Blindness Detection dataset.**

| AFCS-CNN (Model) | Step1 | | Step2 | | Step3 | | Step4 | | Step5 | |
|---|---|---|---|---|---|---|---|---|---|---|
| | Epoch | AF | Epoch | AF | Epoch | AF | Epoch | AF | Epoch | AF |
| AFCS-CNN (VGG16) | 1 | ReLU | 11 | SiLU | 16 | ReLU | 28 | Mish | 32 | SiLU |
| AFCS-CNN (VGG19) | 1 | ReLU | 9 | GELU | 15 | ReLU | 23 | Mish | 29 | GELU |
| AFCS-CNN (DenseNet121) | 1 | ReLU | 3 | GELU | 6 | ReLU | 9 | GELU | 14 | Mish |
| AFCS-CNN (DenseNet169) | 1 | ReLU | 4 | GELU | 9 | ReLU | 11 | GELU | 14 | ReLU |
| AFCS-CNN (EfficientNetV2B0) | 1 | Mish | 2 | ELU | 4 | Mish | 6 | ELU | 9 | Mish |
| AFCS-CNN (EfficientNetV2B1) | 1 | Mish | 3 | SiLU | 5 | Mish | 8 | ReLU | 11 | SiLU |

epoch and overfitting occurred. It was determined that when the VGG16 model is used in the AFCS-CNN model structure, overfitting is prevented, and learning occurs in fewer epochs. Thus, in the training performed with the VGG16 model, it was observed that the AF switch during the training process with the AFCS-CNN model structure had a positive effect on the model training process.

In addition, it was observed that when other CNN models discussed in the study were used in the AFCS-CNN model structure, different AF switches occurred at different epochs and the AF switch had a positive effect on the model training process. When the train-validation accuracy and loss graphs shown in Fig. 6 are examined, the models made five AF switches depending on the number of cycles defined at the beginning. In the training of the CNN models used in the AFCS-CNN model structure on the APTOS 2019 Blindness Detection dataset, different AF switches were made at different epochs, and the details are summarized in Table 11.

After the train-validation process, the final performance of the models was tested using the APTOS 2019 Blindness Detection test dataset, which the network had never seen, and the numerical results are given in Table 12. Additionally, Fig. 7 shows the confusion matrix results obtained from the test dataset with each CNN model used in the AFCS-CNN model structure.

According to the results given in Table 12 and Fig. 7, the most successful classification accuracy in the APTOS 2019 Blindness Detection test dataset was achieved by using the DenseNet121 model in the AFCS-CNN model. With the AFCS-CNN (DenseNet121) model, 83.27% accuracy and 0.5704 loss values were obtained in the test dataset. It has been observed that using all CNN models discussed in the study in the AFCS-CNN model structure gives more successful results than using fixed AF in CNN models. Thus, it has been demonstrated that the AFCS-CNN model structure provides an important performance increase in all CNN models discussed in the study.

## Discussion

To determine the performance of the AFCS-CNN model structure, the experimental results performed with many CNN models and many datasets discussed in this study have shown that the proposed AFCS-CNN model structure provides an important performance

**Table 12 Numerical results obtained from models using the APTOS 2019 Blindness Detection test dataset.**

| Model | Metric | AF | | | | | | AFCS-CNN |
|---|---|---|---|---|---|---|---|---|
| | | ReLU | ELU | SELU | Mish | SiLU | GELU | |
| VGG16 | Accuracy | 0.7709 | 0.7381 | 0.7599 | 0.7945 | 0.7945 | 0.7927 | 0.8254 |
| | Loss | 1.0019 | 0.7306 | 0.6178 | 0.6689 | 0.5773 | 0.9456 | 0.4848 |
| | Precision | 0.7589 | 0.7468 | 0.6838 | 0.7846 | 0.8030 | 0.7921 | 0.8196 |
| | Recall | 0.7693 | 0.7390 | 0.7597 | 0.7970 | 0.7923 | 0.7935 | 0.8234 |
| | F1-score | 0.7601 | 0.7308 | 0.7170 | 0.7838 | 0.7975 | 0.7901 | 0.8130 |
| VGG19 | Accuracy | 0.7981 | 0.7727 | 0.7418 | 0.8090 | 0.7545 | 0.8000 | 0.8181 |
| | Loss | 0.5737 | 0.6036 | 0.6501 | 0.6260 | 0.6962 | 0.5305 | 0.5149 |
| | Precision | 0.7925 | 0.7231 | 0.7238 | 0.7961 | 0.7154 | 0.7902 | 0.8132 |
| | Recall | 0.7971 | 0.7744 | 0.7413 | 0.8078 | 0.7532 | 0.8008 | 0.8173 |
| | F1-score | 0.7943 | 0.7368 | 0.7209 | 0.8013 | 0.7081 | 0.7939 | 0.8092 |
| DenseNet121 | Accuracy | 0.8072 | 0.7654 | 0.7690 | 0.7818 | 0.7327 | 0.7854 | 0.8327 |
| | Loss | 0.7971 | 0.6421 | 0.5645 | 0.5828 | 1.0582 | 0.6388 | 0.5704 |
| | Precision | 0.8080 | 0.8246 | 0.7714 | 0.8173 | 0.7720 | 0.7886 | 0.8335 |
| | Recall | 0.8066 | 0.7634 | 0.7687 | 0.7816 | 0.7325 | 0.7872 | 0.8329 |
| | F1-score | 0.8049 | 0.7611 | 0.7607 | 0.7877 | 0.7251 | 0.7817 | 0.8135 |
| DenseNet169 | Accuracy | 0.7836 | 0.6218 | 0.5836 | 0.7290 | 0.7618 | 0.8163 | 0.8254 |
| | Loss | 1.0642 | 1.0636 | 1.2094 | 0.9553 | 0.6982 | 0.7083 | 0.6344 |
| | Precision | 0.7825 | 0.6644 | 0.6388 | 0.7760 | 0.7766 | 0.8182 | 0.8258 |
| | Recall | 0.7813 | 0.6203 | 0.5853 | 0.7286 | 0.7649 | 0.8196 | 0.8233 |
| | F1-score | 0.7777 | 0.6185 | 0.5693 | 0.7173 | 0.7580 | 0.8027 | 0.8071 |
| EfficientNetV2B0 | Accuracy | 0.6363 | 0.7490 | 0.5690 | 0.7454 | 0.7163 | 0.7127 | 0.8054 |
| | Loss | 1.0799 | 0.8925 | 1.1607 | 1.0322 | 1.5423 | 1.0767 | 0.8327 |
| | Precision | 0.6663 | 0.7050 | 0.6519 | 0.7261 | 0.7260 | 0.6596 | 0.7985 |
| | Recall | 0.6383 | 0.7486 | 0.5675 | 0.7461 | 0.7159 | 0.7113 | 0.8076 |
| | F1-score | 0.6448 | 0.7124 | 0.5953 | 0.7277 | 0.6797 | 0.6634 | 0.8013 |
| EfficientNetV2B1 | Accuracy | 0.6836 | 0.6399 | 0.6454 | 0.7927 | 0.7309 | 0.7363 | 0.8254 |
| | Loss | 1.0460 | 1.2034 | 1.1211 | 1.1672 | 1.0322 | 1.0834 | 0.8308 |
| | Precision | 0.6712 | 0.5901 | 0.6211 | 0.7718 | 0.7080 | 0.7365 | 0.8269 |
| | Recall | 0.6864 | 0.6378 | 0.6473 | 0.7909 | 0.7284 | 0.7374 | 0.8251 |
| | F1-score | 0.6392 | 0.6097 | 0.6188 | 0.7657 | 0.7061 | 0.7077 | 0.8194 |

increase. Experiments with different datasets have demonstrated that using CNN models in the AFCS-CNN model structure is more successful than using fixed AF in CNN models. However, using a CNN model in the AFCS-CNN model structure has an additional runtime cost compared to not using it. Since the hyperparameter values of the AFCS-CNN model are $p_0=3$, $p_1=2$ and $AFCS\_loop=5$, the additional running time of the model will be 11 epochs according to the formula given in Eq. (1). Considering that the patience value in a CNN model is usually given as 5 or 10, the additional runtime cost between 1 and 6 epochs brought by the AFCS-CNN model can be ignored when the performance achieved is taken into account.

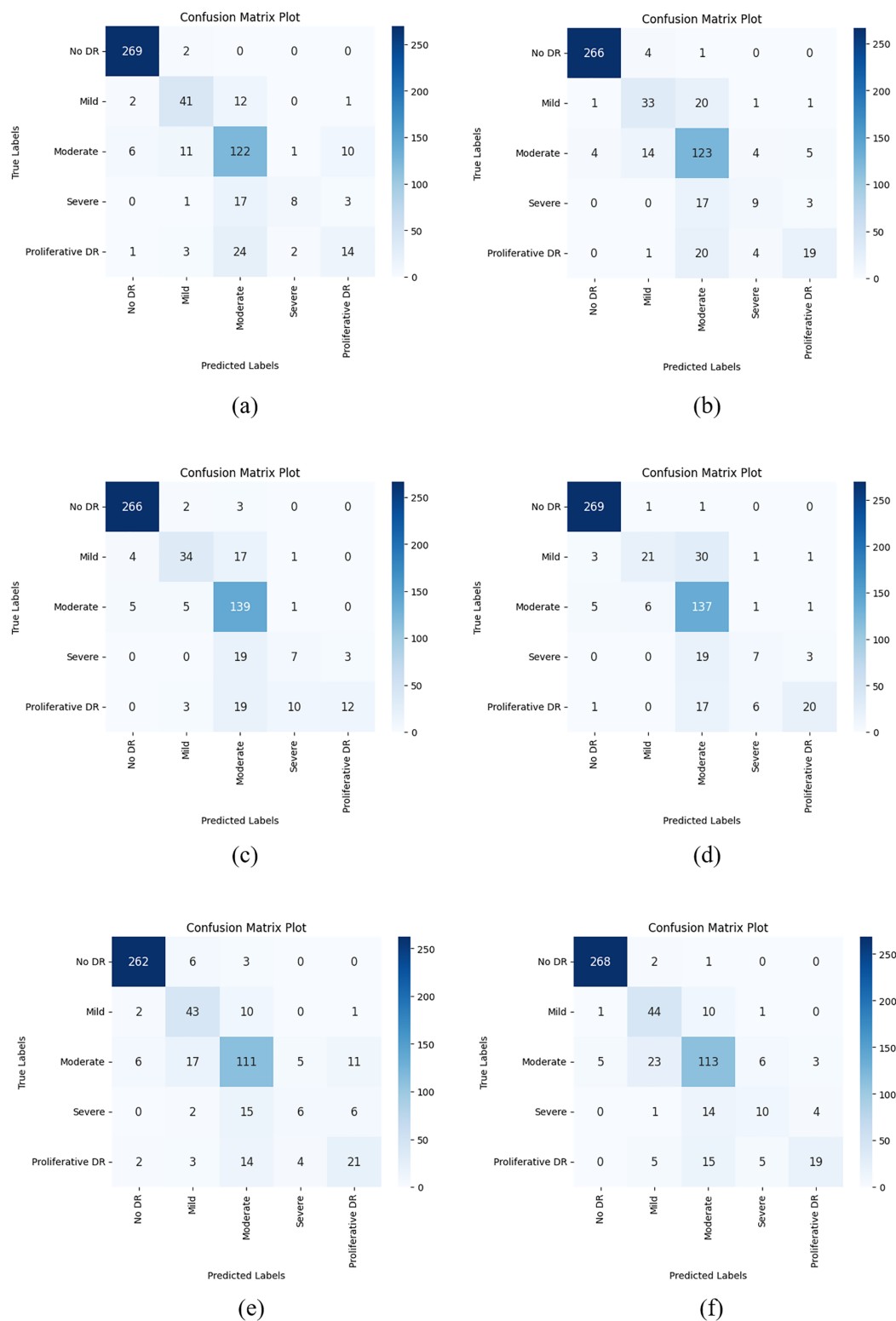

**Figure 7 Confusion matrix plots of each CNN model used in the AFCS-CNN model structure on the APTOS 2019 Blindness Detection test dataset: (A) AFCS-CNN (VGG16), (B) AFCS-CNN (VGG19), (C) AFCS-CNN (DenseNet121), (D) AFCS-CNN (DenseNet169), (E) AFCS-CNN (EfficientNetV2B0), (F) AFCS-CNN (EfficientNetV2B1).**

**Table 13 Comparison of the proposed work with recent works performed on the V2 plant seedling dataset.**

| Reference | Method | Epoch | Accuracy (%) | F1-score (%) |
|---|---|---|---|---|
| *Binguitcha-Fare & Sharma (2019)* | ResNet101 | 500+ | 96.04 | 95.72 |
| *Gupta, Rani & Bahia (2020)* | ResNet50 | 11 | 95.23 | 95.00 |
| *Rahman, Hasan & Shin (2020)* | ResNet-50 | 150 | 96.21 | 95.42 |
| *Fountsop, Ebongue Kedieng Fendji & Atemkeng (2020)* | VGG16 | 100 | 94.24 | – |
| *Makanapura et al. (2022)* | EfficientNetB0 | 50 | 96.52 | 96.26 |
| *Mu et al. (2022)* | Faster R-CNN-FPN | 1,500+ | 95.61 | 91.24 |
| *Tiwari et al. (2023)* | Weed-ConvNet | 50 | 94.20 | 64.00 |
| *Zhang et al. (2023)* | Improved MobileNetV1 | 30 | 96.63 | – |
| *Aliouat, Badis & Bouchiba (2023)* | GA based Automated CNN | 50 | 97.74 | 97.83 |
| Proposed work | AFCS-CNN (DenseNet121) | 21 | 97.11 | 97.21 |

Among the datasets discussed in the study, only the V2 Plant Seedling dataset was compared with current works because various preprocessing was applied to this dataset. For this purpose, to compare the plant seedling classification success of the AFCS-CNN model structure, recent works using the V2 Plant Seedling dataset were examined in the literature and the results are given in Table 13.

When Table 13 is examined, the proposed AFCS-CNN (DenseNet121) model is more successful in plant seedling classification than all other works except the work done by *Aliouat, Badis & Bouchiba (2023)*. Data augmentation was done in the work by *Aliouat, Badis & Bouchiba (2023)*. However, data augmentation was not used in our work. In addition, high success was achieved with the proposed AFCS-CNN models with fewer iterations. Therefore, it was observed that the proposed AFCS-CNN model structure is more successful and competitive with other methods in plant seedling classification. Thus, the AFCS-CNN model structure has achieved state-of-the-art success.

## CONCLUSION

In this study, a new model structure called AFCS-CNN was proposed, which allows cyclical switching of the AF during training, depending on the performance decrease in the neural network, and can be applied to any CNN model. Many experiments were applied to determine the hyperparameters of the proposed model and measure its performance. After a series of ablation studies using the Cifar-10 dataset, hyperparameter values of the AFCS-CNN model structure were determined. In addition, CNN models were chosen to be used in ablation studies. After ablation studies, expansion experiments were performed with V2 Plant Seedling and APTOS 2019 Blindness Detection datasets using VGG, DenseNet, EfficientNetV2, and ConvNeXt versions in the AFCS-CNN model structure.

As a result of expansion experiments, more successful results were obtained with the AFCS-CNN model structure than all the CNN models discussed in the study in both V2 Plant Seedling and APTOS 2019 Blindness Detection datasets. In all experiments, the most successful results were achieved with the AFCS-CNN (DenseNet121) model. Especially in plant seedling classification, a high-test success accuracy of 97.11% was achieved with the

AFCS-CNN (DenseNet121) model with few iterations. In addition, thanks to the use of all other CNN models discussed in the study in the AFCS-CNN model structure, higher test success accuracies were achieved in both datasets. Moreover, the proposed AFCS-CNN model structure was compared with similar works in the literature on plant seedling classification and was more successful than the state-of-the-art works in terms of accuracy and number of iterations. It is planned to use different CNN models in the AFCS-CNN model structure in future studies. Moreover, the applicability of the proposed method in different areas will be tested.

### Funding

The authors received no funding for this work.

### Competing Interests

The authors declare that they have no competing interests.

### Author Contributions

• İsmail Akgül conceived and designed the experiments, performed the experiments, analyzed the data, performed the computation work, prepared figures and/or tables, authored or reviewed drafts of the article, and approved the final draft.

### Data Availability

The V2 Plant Seedlings Dataset is available at Kaggle: https://www.kaggle.com/datasets/vbookshelf/v2-plant-seedlings-dataset.

The APTOS 2019 Blindness Detection Dataset is available at Kaggle: https://www.kaggle.com/c/aptos2019-blindness-detection/data.

### Supplemental Information

Supplemental information for this article can be found online at http://dx.doi.org/10.7717/peerj-cs.2756#supplemental-information.

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
