# Peer review of "Activation function cyclically switchable convolutional neural network model"

_PeerJ Computer Science, doi:10.7717/peerj-cs.2756_

## Round 0.1 · original submission · Major Revisions

Dear authors,

Thank you for your submission. Feedback from the reviewers is now available. Your article has not been recommended for publication in its current form. However, we do encourage you to address the concerns and criticisms of the reviewers and resubmit your article once you have updated it accordingly. Reviewer 3 has asked you to provide specific references. You are welcome to add them if you think they are useful and relevant. However, you are under no obligation to include them, and if you do not, it will not affect my decision.

Best wishes,

Reviewer 1 ·

Basic reporting

The research is interesting, however, this paper is suggested to be published in PeerJ after authors have addressed the comments listed below:

1. For the literature review, authors should refer to more recent research. The current references are not up to date. Authors need to refer to more ISI/Scopus research work instead of the online/conference resources.
2. The literature review need to be further improve. The current review is too general and lengthy. The literature review need to be more specific and need to be tie back to the domain of the research. Please provide a summary at the end of the section to summary the research insights.
3. Authors need to identify the research gap and highlight the contribution(s) of the research work in the manuscript to shows the novelty of the research. The proposed methodology existed in the literature, please highlight the novelty of the research work in the manuscript.
4. Authors should compare the proposed techniques with the state-of-arts [Using the same setting] to further prove the contribution(s)/novelty of the research work. and also to ensure the fairness of the comparisons.
5. More scientific reasoning should be added in the experimental results' explanations.

Experimental design

Refer to the comments in Section 1.

Validity of the findings

Refer to the comments in Section 1.

Additional comments

Refer to the comments in Section 1.

Cite this review as

Reviewer 2 ·

Basic reporting

Following are my observations on this paper.
1. Paper needs to improve
2. Main contributions of the paper is not clearly mentioned but vaguely mentioned.
3 Section 2.2 is not written in english.
4. Any good paper must have related work mentioned in it. Related work is missing in this paper.
5. Mechanism to deternine/identify the best AFs for a particular application is not given. ie how to choose the best AF suitable for a problem in hand is missing here.
6. Why do you call special hyperparameters? How does it is related mathematically with patience hyperparameter.
7. Over all this paper is poorly organised.

Experimental design

1. Mechanism to deternine/identify the best AFs for a particular application is not given. ie how to choose the best AF suitable for a problem in hand is missing here.
2. Why do you call special hyperparameters? How does it is related mathematically with patience hyperparameter.
3. Proposed model in figure 2 needs refinement keeping in mind above 1 and 2 points
4. Model could have been tested in a variety of datasets to arrive at a meaningful conclusion.
5. Mechanism to deal with overfitting, underfitting, high dimensional and highly imbalanced datasets must be included.

Validity of the findings

1. Result analysis section is weak. Comparision with existing approaches must be given.
2. Accuracy Precision Recall, F1 Score etc must be given using table. Only graphical plots will not be convincing to the readers.

Additional comments

1. Technical contributions is weak.
2. Paper needs to improve.

Cite this review as

Reviewer 3 ·

Basic reporting

1. The paper requires further proofreading to enhance the quality of English, as there are several grammatical errors and instances of non-English text. For example, the title of Section 2.2 should be revised, as the current version appears to be in a language other than English.

2. The concept of cyclically switching settings during model training has been introduced. The author should consider discussing related cyclic-switching strategies, such as employing cyclic learning rates [1] and cyclic precision [2] during training.


[1] Smith, Leslie N. "Cyclical learning rates for training neural networks. arXiv." Preprint at https://arxiv. org/abs/1506.01186 (2015).

[2] Fu, Yonggan, et al. "Cpt: Efficient deep neural network training via cyclic precision." arXiv preprint arXiv:2101.09868 (2021).

Experimental design

The conducted experiments provide preliminary evidence of the effectiveness of the proposed solution, though the rigor of the paper would be enhanced by a larger-scale evaluation.

Validity of the findings

The primary concern relates to the generalization ability of the proposed method. Since the author selected the optimal model using CIFAR-10 and then trained and tested it on a different dataset, it raises questions about how well this approach will generalize to other models. If the proposed solution is only applicable to a specific model, its impact will be limited.

Another concern is the efficiency of the proposed method. The approach involves changing activation functions based on model performance testing. When considering the cost associated with this, how does the overall training cost compare to other baseline solutions?

Additional comments

I have a follow-up question for the experiment design. As multiple activation functions are adopted during training, would this approach somehow benefit the robustness of the proposed method? If validated, this could be another strength of the proposed solution.

Cite this review as

---

## Round 0.2 · accepted · Accept

Dear Author,

I would like to express my gratitude for the revised paper. It is worth noting that two of the reviewers did not respond to the invitation to review the revision. However, one reviewer has accepted the paper. I have also conducted my own assessment of the revision. The paper appears to have undergone sufficient improvement, and I am content with the current version. Your revised paper manuscript seems ready for publication.

Best wishes,

Reviewer 3 ·

Basic reporting

The author has addressed my concerns in this part of the previously submitted version.

Experimental design

The author has addressed my concerns in this part of the previously submitted version.

Validity of the findings

The author has addressed my concerns in this part of the previously submitted version.

Cite this review as